# Autoregression with Self-Token Prediction

**Dengsheng Chen** [1 2]   **Yangming Shi** [1 2]   **Enhua Wu**[* 1 2 3]

## Abstract

Conventional autoregressive models achieve causality through next-token prediction, but suffer from prohibitive latency and typically underperform non-causal alternatives such as masked prediction and diffusion. We propose **self-token prediction**, which enables predicting a flexible number of tokens per step, and introduce SAR, the first spatially autoregressive image generator built on this paradigm. SAR delivers markedly faster inference speeds and consistently outperforms prior autoregressive baselines, achieving performance on par with state-of-the-art non-causal models. Our findings highlight self-token prediction as a crucial step toward a high-quality autoregressive paradigm for visual generation.

## 1. Introduction

Autoregressive (AR) modeling via next-token prediction (Vaswani et al., 2017; Radford et al., 2018) has been a central engine behind the success of large language models and physical AI (Wu et al., 2026). Its simplicity, stable training dynamics, and scalability (Kaplan et al., 2020; Henighan et al., 2020) have also made it a natural default choice for multimodal systems (Bordes et al., 2024). In particular, by leveraging pretrained text–vision aligned tokenizers (*e.g.*, CLIP (Radford et al., 2021) and SigLIP (Tschannen et al., 2025; Zhai et al., 2023)) to encode visual data into token sequences, AR models can ingest images and videos as contextual inputs for text generation, achieving impressive results in multimodal understanding and comprehension (Bai et al., 2023; Chen et al., 2022; Ye et al., 2023).

However, when moving from understanding to *multimodal generation*, the next-token paradigm becomes a primary

---

*Corresponding author. [1]Key Laboratory of System Software (Chinese Academy of Sciences) and State Key Laboratory of Computer Science, Institute of Software, Beijing, China [2]University of Chinese Academy of Sciences, Beijing, China [3]Prof. Emeritus, University of Macau, Mocao, China SAR. Correspondence to: Dengsheng Chen <chends@ios.ac.cn>.

*Proceedings of the $43^{rd}$ International Conference on Machine Learning*, Seoul, South Korea. PMLR 306, 2026. Copyright 2026 by the author(s).

bottleneck (Chen et al., 2024c). By design, next-token prediction emits exactly one token per step, which leads to prohibitive inference latency for high-dimensional modalities whose dense spatial or spatiotemporal structure induces extremely long token sequences. Moreover, serializing 2D images (or 3D videos) into a rigid 1D order (such as raster scan) often yields an unnatural generation process that struggles to reflect intrinsic visual structure. These limitations help explain a persistent empirical gap: conventional causal AR models (*i.e.*, VQGAN (Esser et al., 2021)) are typically slower at inference and often underperform strong *non-causal* alternatives, such as masked prediction (Li et al., 2024) and diffusion models (Peebles & Xie, 2023; Nichol & Dhariwal, 2021), which can leverage bidirectional context or iterative denoising to achieve superior sample quality at comparable compute.

Recent works (Santilli et al., 2023; Robbins, 2025; Liu et al., 2024; Stern et al., 2018) have explored engineering optimizations to partially parallelize decoding, yet they largely remain confined to the next-token constraint and therefore do not resolve the root cause: *causality is enforced by generating a single token per step*. In this work, we challenge this coupling and ask a simple question: can we preserve strict (spatial) causality without being bound to next-token prediction?

We answer in the affirmative by introducing **self-token prediction**, a new paradigm for autoregressive generation that generalizes next-token prediction in a principled way (Sec. 2). Instead of being constrained to one token, self-token prediction enables the model to predict a *flexible number of tokens* at each autoregressive step. Crucially, causality is preserved not by unit-step decoding, but by injecting and coordinating dependencies at the *group* level through *group causal attention*. This framework also recovers conventional autoregressive modeling as limiting cases: with a strictly shift-aligned step input, self-token prediction reduces to standard next-group autoregression, where training no longer needs duplicated token streams and inference avoids cache inconsistency, eliminating cache replacement or recomputation. In the extreme case where each group contains a single token, it further collapses to classical next-token prediction. Together, these reductions clarify that self-token prediction is a modular generalization that can match standard AR efficiency when alignment holds, while

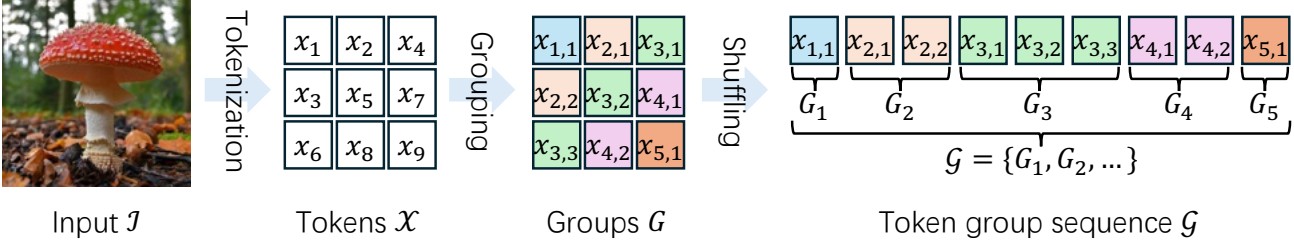

*Figure 1.* **Token grouping pipeline.** An input token sequence $\mathcal{X}$ is partitioned into an ordered set of groups $\mathcal{G} = \{G_t\}_{t=1}^{T}$, where each group $G_t$ is generated in parallel within a single autoregressive step, and the group order defines the causal dependency across steps.

enabling much faster generation via larger token groups when high-dimensional modalities demand it (App. C).

Building on this paradigm, we present SAR, the *first spatially autoregressive image generator* based on self-token prediction (Sec. 3). To make strict spatial causality practical in training, we further examine a set of subtle optimization issues that it can induce, and incorporate several complementary design choices to address them. Specifically, we condition AdaLN-Zero (Peebles & Xie, 2023) on the normalized token-group index so that the model can associate each autoregressive step with its own feature space, which stabilizes optimization and supports continued convergence (App. E). In addition, we add random noise with randomly sampled strength to image tokens to weaken spurious dependencies arising from strong local similarity between nearby tokens, improving the effectiveness of learning under causal factorization (App. 3.4). Finally, we factorize conventional classifier-free guidance (Ho & Salimans, 2022) into $z$-**CFG** and $x$-**CFG**, which helps mitigate the context leakage that can emerge from causal dependencies during training and, in turn, prevents the unconditional branch from being inadvertently compromised (Sec. 3.2). With these improvements, SAR substantially reduces inference latency, consistently outperforms prior autoregressive baselines, and matches the performance of state-of-the-art non-causal models under comparable settings.

Our extensive experiments yield three pivotal insights. First, we demonstrate that the efficacy of autoregressive modeling derives from establishing effective dependency structures rather than strict adherence to single-token prediction. While next-token prediction is well-suited for 1D text, we show it is not inherently optimal for high-dimensional modalities, and that strictly causal generation can remain competitive when causality is enforced via properly designed group dependencies (Sec. 2.5). Second, self-token prediction generalizes robustly across diverse representation spaces (image tokenizer), ranging from continuous latent spaces to raw pixel space. Its strong performance in raw pixel space, in particular, points to a promising avenue for simplifying and unifying multimodal architectures without

relying on any image tokenizer (Sec. 3.3). Third, our method exhibits favorable scaling laws than existing autoregressive models in image generation, establishing it as a highly scalable solution for next-generation multimodal foundation models that require both strict causality and efficient generation (Sec. 4.2).

**Conflict of Interest Disclosure** The authors declare that they have no financial or other substantive conflicts of interest that could reasonably be perceived to influence this work.

## 2. Autoregression with Self-Token Prediction

For images and videos, causality is naturally defined over a *spatial* (or spatiotemporal) partial order rather than a 1D token index. The central goal of self-token prediction is to retain strict causality under such a 2D (or 3D) constraint while reducing autoregressive depth by predicting multiple tokens per step. Throughout this section, we treat the token grouping $\mathcal{G}$, the constructor $h_t(\cdot)$, and the attention mask as the three components that jointly instantiate a spatially causal factorization: grouping determines which positions are generated together, $h_t(\cdot)$ specifies how causally valid information is injected into the current step, and masking enforces that no token attends to spatially "future" positions.

### 2.1. Token Grouping

Let an input instance $\mathcal{I}$ from an arbitrary modality (*e.g.*, an image or a video) be tokenized into a sequence of tokens

$$\mathcal{X} = \{x_1, x_2, \ldots, x_N\}, \tag{1}$$

where each token $x_i$ represents either a discrete code or a continuous feature vector. We partition this sequence into an ordered collection of token group sequence

$$\mathcal{G} = \{G_1, G_2, \ldots, G_T\}, \tag{2}$$

where each token group

$$G_t = \{x_{t,1}, x_{t,2}, \ldots, x_{t,|G_t|}\} \tag{3}$$

is predefined and corresponds to a single autoregressive generation step.

At step $t$, all tokens in $G_t$ are generated in parallel and are treated symmetrically, *i.e.*, we do not impose an intrinsic ordering or causal dependency among tokens within the same group. We place no restrictions on the group cardinalities $|G_t|$, nor do we require groups to have equal sizes across steps. The ordering of groups $(G_1, \ldots, G_T)$ induces a strict causal structure over the token group sequence $\mathcal{G}$: each group $G_t$ is generated conditioned only on the previously generated groups $\mathcal{G}_{<t}$, as illustrated in Fig. 1.

When the underlying data are images (or videos), the group ordering is not merely a convenience: it is the mechanism by which we encode a *spatially causal* partial order. Concretely, we choose a group schedule that respects the desired 2D (or 3D) causality, so that any token position assigned to step $t$ is allowed to depend only on positions assigned to steps $< t$.

## 2.2. Self-Token Prediction

Self-token prediction models the joint distribution over token groups in an autoregressive fashion. At step $t$, the model predicts the ground-truth tokens in $G_t$ conditioned on the previously generated groups $\mathcal{G}_{<t} = \{G_1, \ldots, G_{t-1}\}$.

To form the model input for predicting the current group, we introduce a group-wise input constructor,

$$\tilde{G}_t = h_t(\mathcal{G}_{<t}), \tag{4}$$

where $h_t$ produces one input feature per token position in $G_t$. Conceptually, $h_t$ acts as a causal bridge (or dependency injection): it transfers causally valid information from $\mathcal{G}_{<t}$ into a position-aligned step input $\tilde{G}_t$, thereby specifying how cross-step dependencies are propagated under a chosen grouping. In spatial domains, this role is especially critical: $h_t(\cdot)$ determines *which* previously generated spatial regions can influence the current region, and *how* that information is aligned to the positions being predicted. The constructor $h_t$ can be deterministic or learnable, may depend on any subset of $\mathcal{G}_{<t}$ (or on none of it), and may output fixed embeddings, transformed features, or dedicated placeholder tokens (refer to App. D for more instances). The only requirement is that the output of $h_t$ is dimensionally compatible with the model's input space.

With this construction, the predictive distribution at step $t$ is defined as

$$p_\theta(G_t \mid \mathcal{G}_{<t}) = p_\theta(G_t \mid \tilde{G}_t, \mathcal{G}_{<t}), \tag{5}$$

where $p_\theta(\cdot)$ is parameterized by the model parameters $\theta$, *i.e.*, the set of learnable weights (and biases) optimized during training. Each token $x_{t,i} \in G_t$ is predicted from the representation at its corresponding input position in $\tilde{G}_t$. In contrast to next-token prediction—which typically

predicts a shifted future token—self-token prediction aligns the prediction target with the current input position. [1]

The Transformer backbone enforces group-level causality via masking. Tokens within the same group interact through full self-attention, enabling unrestricted intra-group dependencies. Across groups, attention is constrained such that representations used to predict $G_t$ may attend to tokens from $\mathcal{G}_{<t}$, but must not access the ground-truth tokens from the current or any future groups, *i.e.*, $\mathcal{G}_{\geq t}$. When groups are chosen to respect a spatial partial order, this mask implements *spatial causality*: a prediction at a spatial location can only attend to locations that are designated as causally prior under the chosen 2D ordering.

## 2.3. Efficient Training

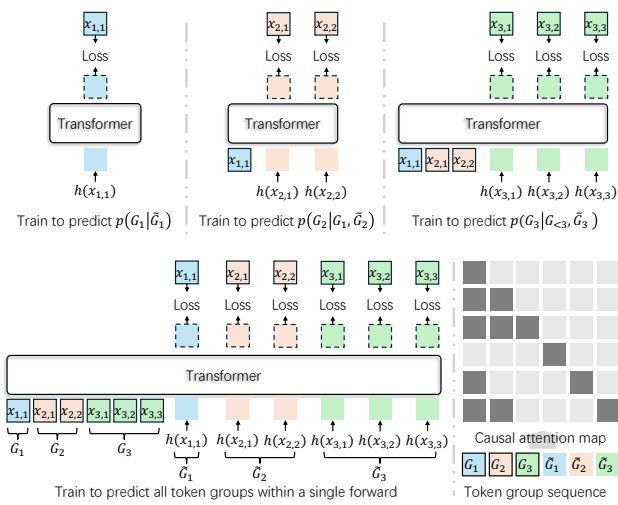

*Figure 2.* **Naïve vs. efficient training pipelines. Top:** a naive implementation predicts one token group per forward pass, which prevents parallel training across groups. **Bottom:** our efficient pipeline predicts all groups in a single forward pass by duplicating each step into a teacher-forced *context* stream and a supervised *generation* stream, with a structured group-causal attention mask that enforces the intended dependencies.

Self-token prediction introduces a training-time inconsistency that complicates optimizing $\theta$ in Eq. 5. The difficulty comes from the dual role of each token group $G_t$: when $G_t$ is the prediction target at step $t$, the model will be driven by a constructed input representation $\tilde{G}_t = h_t(\mathcal{G}_{<t})$; however, when $G_t$ serves as conditioning context for later steps, the model will instead be provided with the ground-truth tokens $G_t$ under teacher forcing. In contrast to standard next-token prediction, where the one-step shift implicitly ties the input stream to the ground-truth prefix, self-token prediction does not constrain $h_t(\cdot)$ to reproduce the ground-truth conditioning stream. This mismatch is particularly pronounced under spatial causality, where the constructor may aggregate or

---

[1]That's why we call it as self-token prediction.

transform information from a spatial neighborhood rather than simply shifting the token stream. As a result, a single representation cannot simultaneously satisfy both roles, which precludes naïve parallel training over multiple groups (Fig. 2, top).

We address this by introducing two aligned copies for every group. The first copy is a *context* group, which directly uses the ground-truth tokens $G_t$ as input and incurs no loss. The second copy is a *generation* group, denoted $\tilde{G}_t$, which uses the constructed input $h_t(\mathcal{G}_{<t})$ and is supervised to predict the ground-truth $G_t$. The two copies share identical positional encodings and differ only in their input embeddings. During training, we concatenate all context groups before all generation groups, *i.e.*, $[G_1, \ldots, G_T, \tilde{G}_1, \ldots, \tilde{G}_T]$, and apply a structured attention mask so that each generation group $\tilde{G}_t$ attends to all context groups in $\mathcal{G}_{<t}$ and to itself, while being prevented from attending to any context tokens in $\mathcal{G}_{\geq t}$ as well as to other generation groups. This mask enforces the intended group-wise causal dependency (and thus the intended spatial causality) without leaking the ground-truth tokens of the current or future steps (Fig. 2, bottom).

Under this construction, the training objective becomes

$$\mathcal{L}(\theta) = \sum_{t=1}^{T} \mathbb{E}\Big[ \ell\big(p_\theta(G_t \mid \mathcal{G}_{<t}, \tilde{G}_t), G_t\big) \Big], \qquad (6)$$

where $\ell(\cdot)$ is task-dependent—for instance, cross-entropy for discrete tokens or a diffusion objective for continuous representations. This formulation allows all groups to be trained simultaneously in a single forward pass. Although the presence of both context and generation streams roughly doubles the number of processed tokens during training, the overhead occurs only when a modality is explicitly trained as a generation target; if a modality serves purely for conditioning, the corresponding generation groups are omitted and no additional computation is required. Under certain conditions, even when used for generation, this extra overhead can be avoided (see App. C for details).

### 2.4. Autoregressive Sampling

At inference time, token groups are generated sequentially. At step $t$, the model predicts $\hat{G}_t$ conditioned on previously generated groups $\hat{\mathcal{G}}_{<t}$, using the constructed input $\tilde{G}_t = h_t(\hat{\mathcal{G}}_{<t})$. As in standard autoregressive decoding, we employ key-value caching to avoid recomputing attention over past groups.

After $\hat{G}_t$ is produced, an inconsistency must be resolved before proceeding: the cached states created during the prediction of step $t$ correspond to the embeddings of $\tilde{G}_t$, whereas subsequent steps must condition on the generated tokens $\hat{G}_t$ as the effective context. We therefore perform

*Table 1.* **Ablation on grouping strategies and causal dependency injection for self-token prediction.** We compare different spatial grouping schemes and input constructors $h(\cdot)$, reporting the resulting number of autoregressive steps, relative training cost, and sample quality (FID-50K). **Training cost.** "$\times 2$" denotes the unavoidable computational overhead incurred during training due to the additional token group for generation (see Sec. 2.3). **Latency.** We report the average wall-clock time measured on a single NVIDIA A100 GPU. All metrics are obtained with SAR-L trained for 250K iterations; sampling uses a flow-matching head with 50 denoising steps.

| Method Configuration | | Efficiency & Cost | | Generation Quality | |
| --- | --- | --- | --- | --- | --- |
| Grouping Strategy | $h(\cdot)$ | #Steps↓ | Train Cost↓ | FID50k↓ | Latency↓ |
| Next-token (Fig. 8a) | $G_{t-1}$ | $h \times w$ | $1\times$ | 4.92 | 622ms |
| Next-row (Fig. 8b) | $G_{t-1}$ | $h$ | $1\times$ | 22.78 | 66ms (9.4$\times$) |
| Next-layer (Fig. 8c) | $\mathrm{Agg_{NN}}(\cdot)$ | $\frac{\max(h,w)}{2}$ | $2\times$ | 29.33 | 45ms (13.8$\times$) |
| Next-nbr (Fig. 8d) | $\mathrm{Agg_{SE}}(\cdot)$ | $h+w-1$ | $2\times$ | 58.49 | 106ms (5.8$\times$) |
| Next-nbr (Fig. 8d) | $\mathrm{Agg_{AVG}}(\cdot)$ | $h+w-1$ | $2\times$ | 4.40 | 106ms (5.8$\times$) |

cache reconciliation (a form of context grounding): we discard the KV entries associated with $\tilde{G}_t$ and recompute them by feeding $\hat{G}_t$ as input, yielding a cache that is consistent with the conditioning required for future steps. The updated cache is then used for generating step $t+1$, as shown in Fig. 3.

While this strategy introduces an additional computation per group, it does not translate into a proportional increase in end-to-end latency. Specifically, the recomputation of the KV states for $\hat{G}_t$ can be scheduled together with the forward pass that predicts the next group $\hat{G}_{t+1}$, so that most of the recomputation cost is effectively hidden behind the decoding of the next step. In practice, this results in only a small constant overhead: compared to the standard group-wise autoregressive decoder, the pipeline incurs essentially one extra forward pass overall (for bootstrapping/initialization), while the remaining per-group cache updates are amortized within the normal decoding passes. Consequently, the inference-time complexity remains linear in the number of groups $T$, and the observed latency is primarily governed by the group-level autoregressive depth rather than the total number of tokens, as shown in Tab. 1.

### 2.5. Empirical Study

Self-token prediction is motivated by the inefficiency of next-token prediction when generating dense modalities (*e.g.*, images) with long token sequences. By allowing flexible group sizes, it reduces the number of autoregressive steps $T$. The key question is *whether the effectiveness of strictly causal autoregressive modeling comes from the next-token factorization itself, or more fundamentally from how causally valid (spatial) dependencies are injected and structured across steps through the constructor $h(\cdot)$.*

To decouple factorization granularity from dependency modeling, we evaluate self-token prediction under three controlled configurations. First, a strict next-token baseline

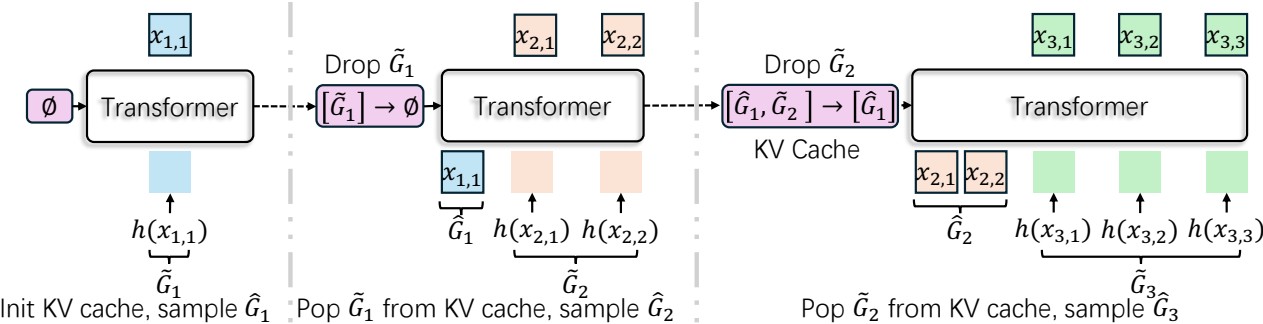

**Figure 3. Sampling with cache grounding.** At step $t$, the model predicts $\hat{G}_t$ from the constructed input $\tilde{G}_t = h_t(\hat{\mathcal{G}}_{<t})$ using key–value caching. Since subsequent steps must condition on the generated tokens, we discard the KV entries corresponding to $\tilde{G}_t$ and recompute them from $\hat{G}_t$, ensuring the cache is consistent for generating $\hat{G}_{t+1}$.

based on raster order sets $|G_t| = 1$ (Fig. 8a) and uses a shift-aligned constructor that returns the ground-truth predecessor, *i.e.*, $\tilde{G}_t = G_{t-1}$. Second, a context-ablated group-wise variant forms groups using Manhattan ordering (Fig. 8d) but intentionally removes informative context by setting the constructed input to a learnable shared embedding, thereby severing the dependency path from $\mathcal{G}_{<t}$ to the current prediction. Third, a structured-dependency group-wise variant uses the same grouping but injects context by aggregating features from spatially adjacent, previously generated groups, which explicitly reflects the intended spatial partial order. (Refer to App. D for more experimental details.)

**Main Findings.** Tab. 1 shows that reducing autoregressive depth alone is insufficient for high-quality strictly causal generation, and that the decisive factor is whether the constructor $h(\cdot)$ injects informative spatial dependencies under the chosen partial order. Moving from next-token raster scanning to more aggressive groupings greatly reduces the number of steps and improves latency, for example next-row decoding cuts steps from $h \times w$ to $h$ and achieves 66 ms (9.4×), and next-layer further reduces steps to $\max(h, w)/2$ with 45 ms (13.8×), yet both suffer severe quality drops (FID 22.78 and 29.33). In contrast, Manhattan neighbor grouping reaches low latency (106 ms, 5.8×) while preserving fidelity, but only when $h(\cdot)$ provides meaningful conditioning: replacing a non-informative shared embedding $\text{Agg}_{\text{SE}}$ (Eq. 17) (FID 58.49) with neighborhood averaging $\text{Agg}_{\text{AVG}}$ (Eq. 18) yields strong samples (FID 4.40), even though the grouping, step count ($h+w-1$), and training cost (2×) remain identical. Together with the observation that the next-token baseline attains FID 4.92 but is bottlenecked by latency (622 ms), these results support the motivation that effective spatial causality is not inherently tied to next-token prediction, but depends on how causal dependencies are structured and injected through $h(\cdot)$ while reducing the autoregressive depth.

## 3. Spatially Autoregressive Image Synthesis

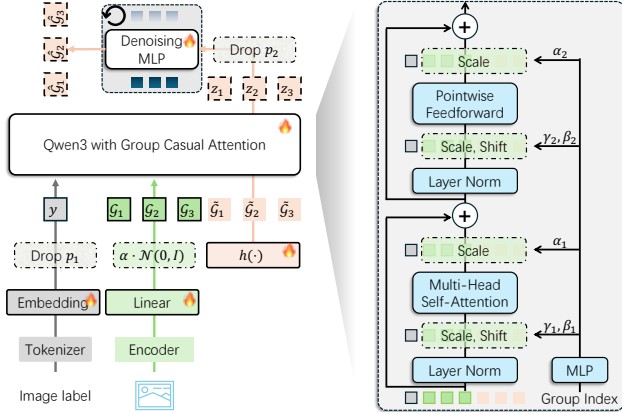

**Figure 4. Overall architecture of SAR.**

This section presents the full methodology of SAR, a native autoregressive image generator built upon *self-token prediction* under strict *spatial causality*. SAR synthesizes continuous visual tokens using a group-wise causal mechanism: token groups are generated sequentially according to a 2D partial order (Fig. 8d), conditioned on all preceding groups and an optional class token, while tokens within the same group are predicted in parallel.

### 3.1. Architecture and Causal Factorization

**Backbone.** SAR builds upon Qwen3 (Yang et al., 2025) and incorporates adaLN-Zero (Peebles & Xie, 2023) to more effectively inject autoregressive timestep signals into the backbone (App. E). The model is conditioned on both the token-group index (*i.e.*, the autoregressive step) and the class label. Concretely, each class label $y$ is mapped to a unique token ID and prepended to the input sequence as a global in-context condition. To enable Classifier-Free Guidance (CFG) (Ho & Salimans, 2022) at inference, we adopt a null-label strategy during training: with probability

$p_1 = 0.1$, we replace $y$ with a learnable `<NULL>` token.

Unless otherwise specified, we use SAR-L (Tab. 6), a 28-layer Transformer with hidden dimension 1152. For positional encoding, we apply 1D RoPE (Su et al., 2024) to the class token and 2D MM-RoPE (Yuan et al., 2025) to image tokens.

A denoising flow head (Chen et al., 2024b), implemented as a 6-layer MLP, is attached to the backbone. Its hidden dimension matches the backbone by default, except in raw-pixel experiments where we increase it to 2048 (Sec. 3.3). Following standard multimodal training practice, we compute the loss only over the generation token groups $\tilde{\mathcal{G}}$, while the context groups $\mathcal{G}$ and the class token $y$ are used solely for conditioning. Figure 4 provides an overview.

We further observe that autoregressive training in continuous feature spaces can suffer from information leakage due to local similarity among neighboring features, which hinders optimization. To mitigate this issue, we add i.i.d. noise with random magnitude $\gamma$ to each token feature, reducing such similarity and empirically improving convergence. Detailed analysis is provided in App. 3.4.

**Visual tokenization.** By default, images are encoded into continuous latent features using the KL-VAE from LDM (Rombach et al., 2022) with a spatial downsampling factor of 16. An image $I \in \mathbb{R}^{3 \times h \times w}$ is transformed into a latent tensor $Z \in \mathbb{R}^{C \times \frac{h}{16} \times \frac{w}{16}}$ with $C = 16$. We use patch size 1 on the latent grid, yielding a sequence of $H \times W$ latent tokens where $H = \frac{h}{16}$ and $W = \frac{w}{16}$. A linear layer projects each latent token to the Transformer hidden dimension.

For the raw-pixel setting (Sec. 3.3), we apply a $16 \times 16$ patchify operation directly on RGB images. Each patch is flattened into a 768-dimensional vector and projected to the backbone hidden size via a linear layer.

We instantiate spatial causality via a Manhattan-distance grouping on the $H \times W$ token grid, as illustrated in Fig. 8d. Tokens are assigned to groups by their Manhattan distance to the origin, so that group indices define a strict 2D partial order over spatial positions.

**Dependency injection under spatial causality.** To leverage local spatial context while strictly adhering to group-level causality, we instantiate the dependency injection function in Eq. (15). Let $\tilde{x}_{i,j}$ denote the (optionally augmented) token embedding at position $(i, j)$ after projection. For tokens in the generation stream, the backbone input is constructed by aggregating causally available neighbors:

$$h(\tilde{x}_{i,j}) = \tilde{x}_{i-1,j} + \tilde{x}_{i,j-1} + \frac{1}{\sqrt{2}} \tilde{x}_{i-1,j-1}, \quad (7)$$

where out-of-bound neighbors are zero-padded. Under Manhattan grouping, each neighbor in Eq. (7) belongs to a

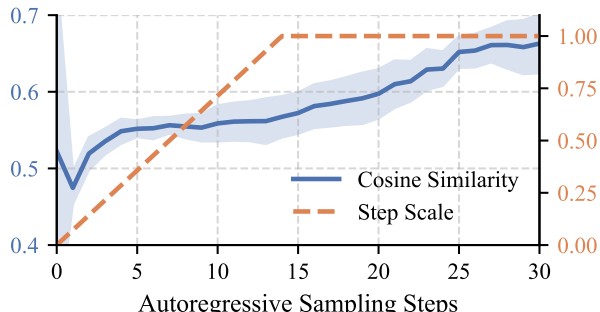

*Figure 5.* **Guidance strength decays with causal context accumulation.** We plot the cosine similarity between the conditional prediction $z_c$ and the guided mixture $z_u + \omega(z_c - z_u)$ across autoregressive sampling steps (blue, left axis), together with the step-wise guidance schedule (orange dashed, right axis). A higher cosine similarity between $z_c$ and $z_u + \omega(z_c - z_u)$ indicates weaker unconditional guidance. As decoding progresses, the conditional and unconditional branches become increasingly similar, which reduces the discrepancy term and weakens the effective guidance at later steps under spatially causal autoregressive generation.

strictly earlier group than $(i, j)$ because $(i-1) + j < i + j$, $i + (j-1) < i + j$, and $(i-1) + (j-1) < i + j$. Therefore, $h(\tilde{x}_{i,j})$ depends only on previously generated groups, ensuring no leakage from the current or future groups. For the context stream, token embeddings are fed directly (after augmentation, if applied).

**Generative objective.** Let $z_{i,j}$ denote the backbone feature at position $(i, j)$, obtained by attending to all causally valid conditioning inputs. The denoising head parameterizes a time-dependent velocity field $v_\theta(\tau, x, z)$ to model conditional token dynamics. We train with a flow-matching objective (Lipman et al., 2022) defined on a linear interpolation path between Gaussian noise and the data distribution:

$$x_{i,j}(\tau) = (1 - \tau) x_{i,j}^{(0)} + \tau x_{i,j}^{(1)}, \qquad \tau \sim \mathcal{U}(0, 1), \quad (8)$$

where $x_{i,j}^{(0)} \sim \mathcal{N}(0, I)$ and $x_{i,j}^{(1)}$ is the target token. This yields a constant target velocity:

$$u_{i,j}(\tau) = \frac{\mathrm{d}}{\mathrm{d}\tau} x_{i,j}(\tau) = x_{i,j}^{(1)} - x_{i,j}^{(0)}. \quad (9)$$

The token-level flow-matching loss is minimized over the generation region $\tilde{\mathcal{G}}$:

$$\mathcal{L}_{\text{FM}} = \mathbb{E}\left[ \frac{1}{\left|\tilde{\mathcal{G}}\right|} \sum_{(i,j) \in \tilde{\mathcal{G}}} \|v_\theta(\tau, x_{i,j}(\tau), z_{i,j}) - u_{i,j}(\tau)\|_2^2 \right]. \quad (10)$$

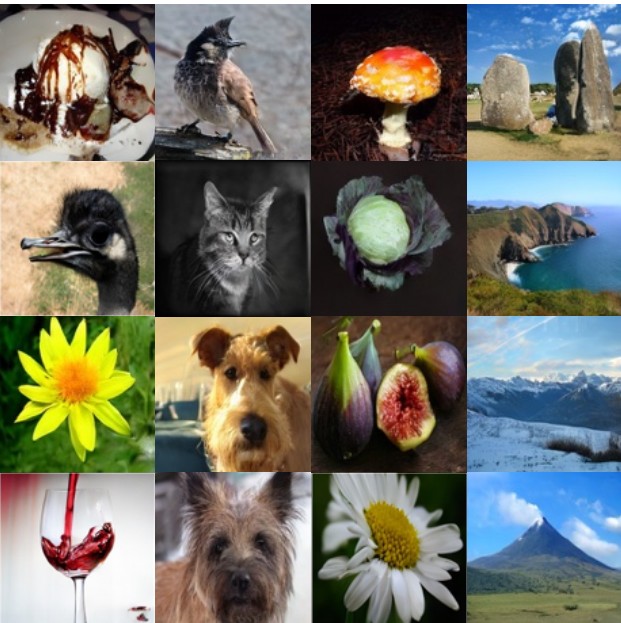

*Figure 6.* **Curated samples from SAR-L on ImageNet-1K** (256 × 256). Images are generated with strictly spatially causal group-wise autoregressive decoding, demonstrating high fidelity and diversity under the proposed self-token prediction paradigm.

We use *v*-prediction (velocity prediction) as our default parameterization. For raw-pixel representations, we instead adopt *x*-prediction to improve training stability, following Li & He (2025). See App. H for a detailed discussion of the effects of *x*- *vs*. *v*-prediction.

### 3.2. Token-level CFG

We observe that autoregressive models, such as Llama-Gen (Sun et al., 2024a), Emu3 (Wang et al., 2024) and Janus-pro (Chen et al., 2025), can be unusually sensitive to classifier-free guidance (CFG) under strict spatial causality. A key reason is that, during autoregressive decoding, the causal context grows monotonically: as more token groups

*Table 2.* **Guidance ablation under spatially causal decoding.** We compare standard CFG, $x$-CFG, and $z$-CFG in terms of computation, latency, and sample quality. Guidance is essential for SAR: without guidance, performance is poor (FID 39.68), while standard CFG at scale 3.50 reaches FID 4.40. $x$-CFG provides a strong quality gain with only a minor latency increase, and combining $x$-CFG with $z$-CFG achieves the best FID (3.56) under the same latency as standard CFG.

| CFG Configuration | | | Efficiency & Cost | | Generation Quality | |
|---|---|---|---|---|---|---|
| CFG | $x$-CFG | $z$-CFG | Computation | Latency↓ | FID50k↓ | IS↑ |
| no-CFG | | | $C + c$ | 73 ms | 39.68 | 50.09 |
| - | 1.25 | - | $C + 2 \times c$ | 81 ms | 18.53 | 77.98 |
| 3.50 | - | - | $2 \times C + 2 \times c$ | 100 ms | 4.40 | **307.34** |
| - | - | 3.50 | $2 \times C + c$ | 95 ms | 6.90 | 287.16 |
| - | 1.25 | 2.70 | $2 \times C + 2 \times c$ | 100 ms | **3.56** | 275.13 |

are generated, they become part of the conditioning history for both the conditional and unconditional branches. Consequently, the unconditional branch—despite not receiving the explicit class condition—can still leverage the increasingly informative visual evidence embedded in previously generated tokens, and may thus implicitly infer class-related cues from the shared context.

To quantify this effect, Figure 5 reports the cosine similarity between conditional and unconditional predictions across autoregressive steps. The similarity increases monotonically, indicating that the two branches progressively converge as decoding proceeds. In CFG, the effective guidance signal is proportional to the difference term between branches; therefore, this convergence directly shrinks the guidance residual and makes guidance less effective in late decoding steps.

To reduce implicit class leakage through the causal context, we strengthen the unconditional pathway during training. Besides the standard null-label replacement used for CFG (replacing the class token $y$ with a learnable null token with probability $p_1$), we additionally apply token-level feature dropout inside the denoising head: with probability $p_2$, we set the backbone feature $z$ fed into the head to zero. This creates an explicit unconditional route within the head, discouraging the unconditional branch from relying on autoregressive context to reconstruct class information.

To compensate for the progressive convergence of the two branches at inference, we follow Ma et al. (2025) and adopt a simple step-wise guidance schedule (*step scale*), illustrated by the orange curve in Fig. 5. Rather than using a constant guidance scale across all autoregressive steps, we employ a piecewise-linear schedule that ramps up guidance over the first half of decoding and keeps it fixed thereafter. This schedule offsets the reduced branch discrepancy in later steps and helps maintain an effective guidance signal throughout decoding.

$x$-**CFG and** $z$-**CFG.** Classifier-free guidance (**CFG**) combines unconditional and conditional velocity predictions via $v_{\text{cfg}} = v_{\text{u}} + \omega(v_{\text{c}} - v_{\text{u}})$, where $v_{\text{u}}$ and $v_{\text{c}}$ denote the unconditional and conditional outputs, respectively, and $\omega$ is the guidance scale. We investigate two complementary factorizations of CFG that apply guidance at different stages of the network. $z$-**CFG** performs guidance in the backbone feature space prior to the prediction head. Specifically, given backbone features $z_{\text{u}}$ and $z_{\text{c}}$, we form $z_{\text{mix}} = z_{\text{u}} + \omega_z(z_{\text{c}} - z_{\text{u}})$, and then evaluate the head once using $v_\theta(\tau, x, z_{\text{mix}})$. In contrast, $x$-**CFG** retains the conditional backbone feature $z_{\text{c}}$ but constructs the unconditional branch inside the head by setting the feature input to $z = \mathbf{0}$. The final guided prediction is obtained by applying guidance at the output level. Figure 12 provides a detailed comparison among standard

*Table 3.* **Representation universality across continuous image spaces.** SAR is evaluated on raw RGB patches and VAE latent spaces, including joint image–video latents (WanX (2025)) and KL-VAE latents (2022), under a fixed $16 \times 16$ effective token grid for $256 \times 256$ images. We report FID50k/IS and compare guidance variants. **S&P** denotes the VAE stride (if applicable) and the patch size. Flow matching remains effective across spaces and consistently outperforms a diffusion-style (DDPM) objective under comparable settings.

| Method Configuration | | | CFG Configuration | | | Generation Quality | |
|---|---|---|---|---|---|---|---|
| Denoising Head | Denoising Space | S&P | CFG | $x$-CFG | $z$-CFG | FID50k↓ | IS↑ |
| FM (2022) | Raw (2025) | (-,16) | -
6.50 | 1.50
- | 4.50
- | 8.36
8.42 | 240.36
319.88 |
| FM (2022) | Joint-IV (2025) | (8,2) | -
4.50 | 1.15
- | 3.50
- | 4.84
5.31 | 314.32
342.39 |
| FM (2022) | Latent (2022) | (16,1) | -
3.50 | 1.25
- | 2.70
- | 3.56
4.40 | 275.13
307.34 |
| DDPM (2021) | Latent (2022) | (16,1) | -
12.50 | 1.80
- | 3.90
- | 29.48
16.85 | 140.33
301.25 |

**CFG, $z$-CFG, and $x$-CFG.**

Table 2 shows that disabling guidance causes a substantial degradation in sample quality (FID 39.68 at 73 ms). Standard **CFG** with $\omega = 3.50$ markedly improves fidelity (FID 4.40 at 100 ms) and achieves the best inception score (IS 307.34), indicating that under strict spatial causality guidance is a primary driver of generation quality rather than a minor refinement. Compared to no-CFG, $x$-**CFG** yields a large fidelity gain with only a modest latency increase (FID 18.53 at 81 ms). $z$-**CFG** further improves fidelity while incurring less overhead than standard **CFG** (FID 6.90 at 95 ms). Notably, combining $x$-**CFG** and $z$-**CFG** achieves the best fidelity (FID 3.56) at the same latency as standard **CFG** (100 ms), suggesting that disentangling guidance paths and explicitly preserving an unconditional route is particularly beneficial under causal context accumulation.

### 3.3. Empirical Study

To evaluate the robustness of our spatially causal autoregressive paradigm, we test SAR across multiple continuous representation spaces while keeping a fixed effective token grid of $16 \times 16$ for $256 \times 256$ images. Specifically, we consider raw RGB patch tokens (Li & He, 2025) and latent spaces produced by VAEs, including an ImageNet KL-VAE (Rombach et al., 2022) (Latent) and a joint image–video VAE (WanX) (Wan et al., 2025). We also examine objective flexibility by comparing flow matching (Lipman et al., 2022) against DDPM (Nichol & Dhariwal, 2021).

Table 3 shows that SAR maintains strong performance across these continuous spaces when trained with flow matching. In the KL-VAE latent space, combining $x$-**CFG** and $z$-**CFG** achieves our best result (FID 3.56, IS 275.13), improving upon standard CFG (FID 4.40, IS 307.34). In the raw-pixel setting, SAR remains competitive (FID 8.36, IS 240.36) without relying on any pretrained autoencoder,

indicating that the method does not hinge on a particular latent representation. In contrast, substituting flow matching with a DDPM-style objective substantially degrades fidelity in the same KL-VAE latent space (FID 29.48 with $xz$-**CFG**, and 16.85 with CFG), suggesting that flow matching is a better default objective within our spatially causal autoregressive framework. Additional qualitative results are provided in App. J.

### 3.4. Mitigating Local Shortcut via $\gamma$-Noise

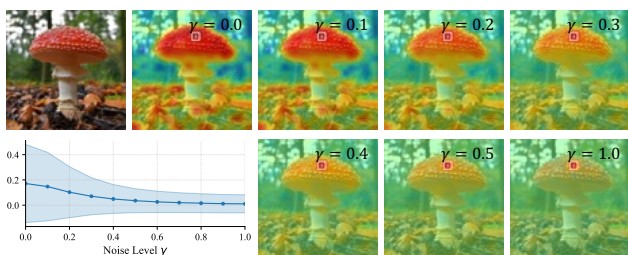

*Figure 7.* **Suppressing similarity-driven attention with noise.** We visualize, for a fixed query token (white box), the cosine-similarity response over all spatial positions computed from continuous image embeddings under different noise levels $\gamma$ in (11). Without noise, the embedding exhibits strong similarity not only to nearby tokens but also to many distant positions, indicating spurious global responses driven by appearance similarity that can interfere with RoPE's distance-aware inductive bias. Increasing $\gamma$ progressively suppresses these responses. **Bottom-left:** average cosine similarity (mean with variation) as a function of $\gamma$, showing a monotonic reduction; $\gamma = 0.5$ already yields a large suppression while remaining compatible with effective training in our experiments.

Unlike discrete autoregressive models where token identities largely avoid feature-level locality artifacts, continuous visual embeddings exhibit strong appearance-driven similarity. Meanwhile, spatial positional encodings such as RoPE explicitly inject distance-dependent structure into the attention computation. In principle, RoPE should help the model organize dependencies by geometric proximity, while the feature space focuses on semantic compatibility. In practice, however, the strong local similarity of continuous image embeddings interferes with this separation. As shown in Fig. 7, a query token not only responds strongly to its spatial neighbors, but also yields unexpectedly high responses to many distant positions. These responses are largely induced by feature similarity rather than semantic relevance, and they effectively blur the distance-aware inductive bias that RoPE is designed to provide. This mismatch results in shortcut learning during training and ultimately harms generation quality.

To suppress such similarity-driven shortcuts, we adopt a simple yet effective augmentation inspired by d-VAE (Sun et al., 2024b) and NextStep-1 (Team et al., 2025). After projecting tokens to the backbone hidden size, we inject

*Table 4.* **Comparison with state-of-the-art models on ImageNet-1K at** $256 \times 256$. We report FID-50K (lower is better) and Inception Score (IS; higher is better). **Spatial Causality** indicates whether the model enforces spatially causal modeling (*e.g.*, disallowing full attention). Notably, although VAR (Tian et al., 2024) is autoregressive, it uses full attention within each resolution stage and thus does not satisfy strict spatial causality.

| Model | Spatial Causality | #Params | #Steps↓ | FID50k↓ | IS↑ |
|---|---|---|---|---|---|
| *GAN* | | | | | |
| BigGAN (Brock, 2018) | ✗ | 112M | 1 | 6.95 | 224.5 |
| GigaGAN (Kang et al., 2023) | ✗ | 569M | 1 | 3.45 | 225.5 |
| StyleGAN-XL (Sauer et al., 2022) | ✗ | 166M | 1 | 2.30 | 265.1 |
| *Diffusion* | | | | | |
| LDM-4-G (Rombach et al., 2022) | ✗ | 400M | 250 | 3.60 | 247.7 |
| DiT-XL/2 (Peebles & Xie, 2023) | ✗ | 675M | 250 | 2.27 | 278.2 |
| JiT-L/16 (Li & He, 2025) | ✗ | 459M | 50 | 2.36 | 298.5 |
| ADM (Dhariwal & Nichol, 2021) | ✗ | 554M | 250 | 10.94 | 101.0 |
| VDM++ (Kingma & Gao, 2023) | ✗ | 2.0B | 512 | 2.12 | 267.7 |
| *Mask Prediction* | | | | | |
| MAGVIT-v2 (Yu et al., 2023) | ✗ | 307M | 64 | 1.78 | 319.4 |
| MaskGIT (Chang et al., 2022) | ✗ | 227M | 8 | 6.18 | 182.1 |
| MAR-L (Li et al., 2024) | ✗ | 479M | 256 | 1.78 | 296.0 |
| *Autoregressive (Next-Token/Scale Prediction)* | | | | | |
| VAR-d20 (Tian et al., 2024) | ✗ | 600M | 10 | 2.57 | 302.6 |
| VQGAN (Esser et al., 2021) | ✓ | 1.4B | 256 | 15.78 | 74.3 |
| ViT-VQGAN (Yu et al., 2021) | ✓ | 1.7B | 1024 | 4.17 | 175.1 |
| RQTran. (Lee et al., 2022) | ✓ | 3.8B | 68 | 7.55 | 134.0 |
| LlamaGen-XXL (Sun et al., 2024a) | ✓ | 1.4B | 256 | 2.34 | 253.9 |
| *Autoregressive (Self-Token Prediction)* | | | | | |
| SAR-B ($\omega_x$=1.35,$\omega_z$=3.0) | ✓ | 141M | 31 | 4.23 | 231.1 |
| SAR-L ($\omega_x$=1.25,$\omega_z$=2.7) | ✓ | 714M | 31 | 2.52 | 288.5 |
| SAR-H ($\omega_x$=1.20,$\omega_z$=2.5) | ✓ | 1.3B | 31 | 2.26 | 290.0 |

bounded Gaussian noise into every token embedding in both the context and generation streams:

$$\tilde{x}_i \leftarrow \tilde{x}_i + \alpha \cdot \epsilon, \quad \alpha \sim \mathcal{U}(0,\gamma), \ \epsilon \sim \mathcal{N}(0,I), \quad (11)$$

where $\gamma$ controls the maximum perturbation magnitude. We then apply the causal constructor $h(\cdot)$ in (7) to these perturbed embeddings. As Fig. 7 illustrates, increasing $\gamma$ markedly reduces the average cosine similarity responses, thereby weakening spurious correlations caused by local appearance similarity. This allows RoPE to more cleanly express distance in attention, while the feature channel is encouraged to focus on higher-level semantic dependencies. Empirically, we find that moderate noise (*e.g.*, $\gamma = 0.5$) substantially suppresses these correlations while still preserving effective learning and improving FID convergence, even though the training loss may increase.

# 4. Experiments

## 4.1. Experimental Setup

**Training details.** We train on ImageNet-1K (Deng et al., 2009) with $256 \times 256$ resolution and random horizontal flips. Following the spatially causal protocol in Sec. 3, we predict token groups in parallel via group-causal attention. We use AdamW (Loshchilov & Hutter, 2017) ($\beta_1 = 0.9, \beta_2 = 0.95, wd = 0.05$) in bfloat16 with a batch size of 1,024. The learning rate is set to $10^{-4}$ after a 5-epoch warmup from $10^{-7}$. An EMA with decay 0.9996 is maintained. Models are trained for 200 epochs for ablations and up to 600 epochs for main results, employing early stopping to

mitigate overfitting.

**Inference and evaluation.** All inference runs use guidance, with the guidance configuration and scale tuned per model variant. For flow-based sampling, we use 30 denoising steps unless otherwise specified. We report Fréchet Inception Distance (FID) (Heusel et al., 2017) and Inception Score (IS) (Salimans et al., 2016), computed from 50,000 generated samples (50 per class) against the full ImageNet-1K validation set, following standard class-conditional evaluation protocols.

## 4.2. Quantitative Comparison

In Tab. 4, we systematically compare representative GAN, diffusion, mask-prediction, and autoregressive (AR) methods. A key advantage of spatially causal methods is their compatibility with KV caching. Consequently, although they require multiple generation steps, the computational overhead is significantly lower than non-causal approaches that necessitate full attention computation at every step. Regarding generation quality, while SAR exhibits a performance gap compared to state-of-the-art mask-prediction and diffusion models at smaller parameter scales, it demonstrates a more pronounced scaling law. This characteristic is paramount in the era of large foundation models, suggesting that performance limitations can be effectively overcome by increasing model parameters. Empirically, although a gap persists against the absolute best methods, SAR-H (FID 2.26) already achieves parity with representative strong baselines like DiT-XL/2 (Peebles & Xie, 2023) (FID 2.27) and significantly outperforms other spatially causal autoregressive counterparts (*e.g.*, LlamaGen-XXL (Sun et al., 2024a)).

## 4.3. Qualitative Results

Figure 6 shows class-conditional samples from SAR-L on ImageNet-1K at $256 \times 256$. The model produces diverse images with coherent global structure and fine-grained details, consistent with the quantitative results. Importantly, these samples are generated under strict spatial causality with group-wise autoregressive decoding, illustrating that parallel token prediction per step can preserve visual fidelity without relaxing causal constraints.

# 5. Conclusion

In this work, we introduce self-token prediction, which enables efficient autoregressive generation for multimodal data while strictly subsuming standard next-token prediction. We further design the SAR and successfully validate the effectiveness of self-token prediction on image generation tasks. Our experiments demonstrate significant advantages in both efficiency and quality when generating dense modalities, establishing a solid foundation for future extensions.

## Acknowledgements

This work is supported by the National Natural Science Foundation of China (NSFC) under Grant No. 62332015.

## Impact Statement

This paper presents SAR, a novel spatially autoregressive image generator based on self-token prediction. From a technological perspective, our work bridges the performance and latency gaps between autoregressive models and non-causal alternatives (*e.g.*, diffusion models) in visual generation. By enabling the prediction of a flexible number of tokens per step, SAR significantly reduces inference latency and computational overhead. This improved efficiency has positive environmental implications by lowering the energy consumption and carbon footprint associated with deploying large-scale generative models. Furthermore, it democratizes access to high-quality image generation, allowing researchers and creators with limited computational resources to leverage state-of-the-art tools. However, like other advanced generative models, our approach carries potential societal risks. The ability to generate high-quality, photorealistic images efficiently could be misused to create deceptive content, such as deepfakes, misinformation, or non-consensual imagery. Additionally, the model may inadvertently inherit and propagate biases present in its training data. To mitigate these risks, we strongly advocate for the responsible use of this technology. We encourage the community to integrate robust safety filters, develop reliable watermarking and AI-generated content detection mechanisms, and carefully curate training datasets to ensure fairness and safety in future deployments.

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

# A. Related Works

**Standard and Multi-Scale Autoregressive Models.** A dominant paradigm for multimodal generation extends large language models by discretizing visual data into token sequences and applying next-token prediction under a fixed spatial order, typically raster-scan (*e.g.*, Chameleon (Team, 2024), Emu3 (Wang et al., 2024), LlamaGen (Sun et al., 2024a)). This design inherits the strengths of AR modeling—simple likelihood factorization and strict causality—but also inherits its most severe limitation for images and videos: the unit-step constraint of emitting exactly one token per decoding step leads to prohibitive inference latency as sequence lengths grow with spatial and temporal resolution. A line of work improves efficiency through structured factorization or multi-scale scheduling. For example, VAR (Tian et al., 2024) alleviates latency via a coarse-to-fine, next-scale prediction strategy, but it remains fundamentally sequential: each step is strictly conditioned on previous scales, and generation progresses through a predefined hierarchy. Such scale-dependent designs also introduce rigid inductive biases tied to 2D layouts, which can complicate extension to other modalities and settings where a single universal hierarchy is undesirable. In contrast, self-token prediction targets the root cause—the next-token constraint itself—by allowing a flexible number of tokens to be predicted per autoregressive step while maintaining strict (spatial) causality through group-level dependency design.

**Masked Signal Prediction.** Masked signal prediction has emerged as a compelling alternative for visual synthesis (Chen et al., 2024a; Li et al., 2023). Approaches such as MaskGIT (Chang et al., 2022) and MAR (Li et al., 2024) partition inputs into masked and unmasked regions and predict missing content from visible context, enabling substantial parallelism during decoding. This efficiency, however, typically relies on bidirectional (full) attention to aggregate global context, which breaks strict spatial causality and prevents the direct use of standard causal key–value caches for efficient incremental decoding. Moreover, the mismatch between random masking during training and iterative refinement at inference often necessitates carefully tuned decoding schedules to avoid quality degradation. Our self-token prediction framework can be viewed as a strictly causal alternative to masked prediction: it retains parallelism by predicting token *groups* per step, yet preserves causal attention via group-causal constraints, keeping decoding compatible with standard AR infrastructure and caching.

**Hybrid and Modular Architectures.** To combine the semantic reasoning capabilities of LLMs with the high-fidelity synthesis of diffusion models, hybrid architectures have been widely explored. These range from tightly coupled systems (*e.g.*, BAGEL (Deng et al., 2025), Transfu-sion (Zhou et al., 2024)) to loosely coupled modular frameworks (*e.g.*, MetaQueries (Pan et al., 2025)). While effective, these approaches often rely on heterogeneous generation mechanisms—autoregressive decoding for text and diffusion-style sampling for images—which introduces disjoint inference procedures (*AR decoding vs. diffusion sampling*), additional system complexity, and nontrivial coordination of objectives, schedules, and compute budgets. Cross-modal interaction is also frequently mediated through shallow fusion or high-level prompting rather than a unified causal generation process. Distinct from this direction, self-token prediction aims to close the quality–efficiency gap *within* a single, strictly causal transformer by reformulating the AR objective to remove the single-token bottleneck, thereby reducing the need for external non-causal backbones.

Overall, existing methodologies are commonly (i) constrained by the sequential bottleneck inherent to next-token prediction, (ii) dependent on non-causal attention for parallelism as in masked prediction, or (iii) reliant on heterogeneous hybrid designs that complicate unified modeling and sampling. Self-token prediction diverges from these paradigms by reformulating autoregressive generation at the objective and dependency-structure level: it enables efficient parallel token emission per step while preserving strict spatial causality.

# B. Scaling Laws of Self-Token Prediction Models

Compared to diffusion transformers (DiT) and mask-based autoregressive generative models—which typically achieve strong ImageNet performance with only a few hundred million parameters and exhibit smooth, monotonic improvements as model scale increases—SAR, a self-token prediction architecture, demonstrates a markedly distinct scaling behavior. Specifically, we observe a pronounced **capacity threshold effect**: models with fewer than approximately 500M parameters consistently fail to learn meaningful representations, yielding near-random generation quality regardless of incremental capacity growth within the 100M–500M range. Strikingly, upon surpassing this critical threshold (e.g., at 700M parameters), performance undergoes an abrupt phase transition, jumping from ineffective to competitive results without intermediate gains. This discontinuous emergence of capability suggests a non-linear scaling regime fundamentally different from the power-law trends observed in conventional generative frameworks.

Moreover, once the capacity threshold is satisfied, further parameter scaling yields diminishing returns unless accompanied by a proportional increase in training data volume, indicating tight coupling between model scale and data requirements. As summarized in Tab. **??**, this behavior

contrasts sharply with DiT and mask-prediction models, which exhibit more graceful scaling. We posit that the self-token prediction paradigm imposes stricter structural demands on model capacity—potentially to sustain stable autoregressive dynamics or capture long-range token dependencies—resulting in a regime where capability emerges abruptly only after crossing a critical architectural scale. This finding underscores that scaling laws are not universal but highly architecture-dependent, with certain generative formulations exhibiting threshold-driven phase transitions rather than smooth scaling trends.

## C. Connections and Limiting Cases

Self-token prediction is a general and modular framework for group-wise autoregressive modeling. This generality subsumes standard autoregressive training and decoding as limiting cases. Importantly, it also makes precise when the additional token groups can be removed without changing the modeled conditional distribution.

**Next-Group Prediction.** Consider the setting where, for a given instance, all token groups have equal cardinality and the input constructor satisfies the strict shift-alignment condition

$$\tilde{G}_t = h_t(\mathcal{G}_{<t}) \equiv G_{t-1}. \tag{12}$$

This induces a one-to-one positional correspondence between tokens in successive groups. Under this condition, the representation that drives prediction at step $t$ is exactly the representation that must be exposed as context for subsequent steps. Consequently, the training-time separation between a ground-truth context stream and a constructed generation stream becomes redundant: token duplication can be eliminated and the model can be trained with a standard group-wise teacher-forcing procedure. At inference time, the same alignment removes cache inconsistency—key-value states computed from the step input remain valid context for future steps—so no cache replacement or recomputation is required. Overall, self-token prediction reduces to conventional group-wise autoregressive modeling with standard decoding efficiency.

**Next-Token Prediction.** A further specialization arises when each token group contains exactly one token, *i.e.*, $|G_t| = 1$ for all $t$. Under the same shift-alignment condition in (12), the group-wise conditional reduces to the standard token-wise autoregressive factorization

$$p(x_t \mid x_{<t}), \tag{13}$$

which is precisely conventional next-token prediction. This establishes next-token prediction as a strict special case of self-token prediction, corresponding to the singleton-group limit with aligned step inputs.

**Next-Scale Prediction.** Self-token prediction is not the only approach that predicts multiple tokens per causal step. A representative example is next-scale prediction (*e.g.*, VAR), which generates from low to high resolutions. Under the self-token prediction view, next-scale prediction corresponds to a particular choice of token groups and input construction: each $G_t$ represents the entire image at a given resolution level, and the constructor is defined by interpolation-based upsampling from the previous resolution,

$$\tilde{G}_t = h_t(\mathcal{G}_{<t}) \equiv \mathrm{Up}(G_{t-1}), \tag{14}$$

where $\mathrm{Up}(\cdot)$ denotes upsampling/interpolation (not differencing). While multi-scale generation is often effective for preserving global structure in synthesis, explicitly modeling intermediate-resolution tokens may introduce objective mismatch in settings where the downstream goal emphasizes representation learning or understanding. In contrast, spatially partitioned self-token prediction keeps groups within a consistent semantic domain and relies on $h(\cdot)$ to propagate structured local-to-global dependencies across groups, avoiding the need to introduce redundant intermediate-scale token sequences.

## D. More Empirical Studies

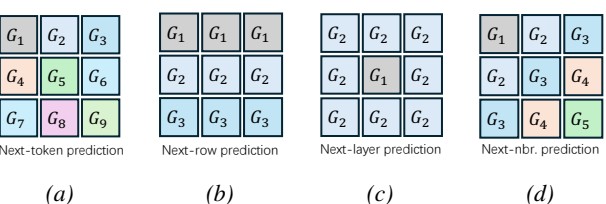

*Figure 8.* **Grouping strategy.** Note that next-token prediction 8a and next-row prediction 8b satisfy the shift-alignment condition in (12), and therefore require no additional training overhead (no context/generation duplication) and no inference-time cache reconciliation.

**Token grouping.** Figure 8 summarizes the token-grouping strategies studied in this paper. Figure 8a illustrates the widely-used raster-scan order, which generates tokens from left-to-right and top-to-bottom, treating each token as a separate group. Figure 8b shows a row-wise variant that generates an entire row per step. Notably, for both raster-scan and row-wise grouping, when we train with teacher forcing by feeding the ground-truth from the previous group(s), the implementation can avoid the extra overhead introduced by maintaining explicit generation groups. Figure 8c groups tokens by expanding outward from the image center, which substantially reduces the number of groups and thus lowers generation latency; however, it often weakens dependency propagation across directions, leading to visible directional artifacts (different directions are insufficiently constrained, causing spatial discontinuities). Finally, Figure 8d adopts

a Manhattan-distance-based grouping, which keeps successive groups spatially adjacent and empirically provides better stability and easier learning of spatial dependencies. In practice, to help the network better accommodate varying image sizes, we start generation from the bottom-right corner (the region with the largest position id) and progressively decode toward the origin; this Manhattan-based schedule is our default at inference time.

**Dependency injection.** To explicitly construct a causally valid input for group $t$, we define a dependency-injection operator based on its causal spatial neighborhood. Let $\mathcal{N}(t)$ denote an index set of causal neighbors of group $t$. We build the injected group input $\tilde{G}_t$ from already-available groups $\mathcal{G}_{<t}$ via

$$\tilde{G}_t = h_t(\mathcal{G}_{<t}) \equiv \mathrm{Agg}(\{G_s : s \in \mathcal{N}(t)\}), \quad (15)$$

where $\mathrm{Agg}(\cdot)$ is a fixed aggregation operator. We instantiate $\mathrm{Agg}(\cdot)$ with three practically relevant forms. (i) Nearest-neighbor propagation selects the closest causal neighbor:

$$\mathrm{Agg}_{\mathrm{NN}}(\{G_s\}) = G_{s^\star}, \quad s^\star = \arg\min_{s \in \mathcal{N}(t)} d(t, s), \quad (16)$$

where $d(t, s)$ denotes the spatial distance between groups (e.g., induced by the Manhattan ordering). (ii) A shared embedding placeholder uses a learned constant embedding, identical across steps:

$$\mathrm{Agg}_{\mathrm{SE}}(\{G_s\}) = E_{\mathrm{shared}}, \quad (17)$$

where $E_{\mathrm{shared}}$ has the same shape as a group input. (iii) Average neighbor aggregation takes the mean over the causal neighborhood:

$$\mathrm{Agg}_{\mathrm{AVG}}(\{G_s\}) = \frac{1}{|\mathcal{N}(t)|} \sum_{s \in \mathcal{N}(t)} G_s. \quad (18)$$

**Video generation.** Finally, although the above empirical study centers on image generation, *self-token prediction* is not specific to images. Its core motivation is to lift the per-token decoding bottleneck of next-token prediction, thereby reducing autoregressive latency on dense modalities. Designing optimal grouping rules and dependency structures for videos or audio-visual signals is beyond the scope of this work; nevertheless, we present a preliminary proof-of-concept on video generation. As shown in Fig. 10, we generate videos autoregressively in a frame-by-frame manner using self-token prediction. Compared to diffusion-based or mask-prediction approaches, a practical distinction is that frames can be rendered online as soon as they are generated, without waiting for the entire video to finish. This property suggests a promising direction for real-time and interactive applications.

*Table 5.* **Ablation study on key components.** We evaluate the effectiveness of AdaLN-Zero, $\gamma$-Noise, Learnable Positional Embedding (Learn-PE), and MM-RoPE. Train for 100k, *i.e.*, 80 epochs.

| AdaLN-Zero | $\gamma$-Noise | Learn-PE | MM-RoPE | FID-50k $\downarrow$ |
|:---:|:---:|:---:|:---:|:---:|
| – | – | – | ✓ | 12.34 |
| ✓ | – | – | ✓ | 10.12 |
| – | ✓ | – | ✓ | 9.45 |
| ✓ | ✓ | – | ✓ | **6.54** |
| ✓ | ✓ | ✓ | – | 7.26 |

**Visual understanding.** Moreover, as discussed in App. C, next-token prediction is a special case of self-token prediction. This connection enables a simple yet appealing training strategy: within the same token sequence, we can jointly optimize self-token prediction and next-token prediction objectives. A key benefit is the potential to build a unified multimodal framework that supports both understanding and generation under a consistent autoregressive formulation. While developing a strong understanding model based on self-token prediction is far beyond the scope of this paper, we conduct an initial exploration by training a lightweight image-and-video understanding model on top of SAR-L. Figure 16 shows qualitative results indicating that, even with minimal additional training, SAR-L can localize spatial regions accurately and produce detailed, contextually grounded descriptions.

# E. Ablations

**Autoregressive steps via adaLN-Zero.** A key design in SAR is to explicitly condition the backbone on the token-group index (*i.e.*, the autoregressive step) through adaLN-Zero (Peebles & Xie, 2023). Concretely, for each token position we associate its group index $g(i, j) \in \{1, \ldots, T\}$ (determined by the chosen spatially causal grouping), embed it, and use the resulting embedding to modulate Transformer blocks via adaLN-Zero. This step-conditioning encourages the model to learn group-specific feature spaces that remain consistent with the evolving causal context across decoding steps. Empirically, it stabilizes optimization and mitigates the tendency of late-stage training to stall under strict spatial causality. As shown in Tab. 5, when MM-RoPE is enabled, adding adaLN-Zero alone improves FID-50k from 12.34 to 10.12. Furthermore, to strengthen class supervision on ImageNet, we add the class embedding to the step embedding and use their sum as the conditioning signal. Although not strictly required, we find this practice can further stabilize training while introducing negligible computational overhead.

$\gamma$-**Noise.** With MM-RoPE fixed, introducing $\gamma$-Noise substantially improves performance, reducing FID-50k from 12.34 to 9.45 (Tab. 5). Beyond this standalone gain, $\gamma$-Noise is complementary to adaLN-Zero: enabling both achieves the best result, reaching an FID-50k of 6.54. We attribute

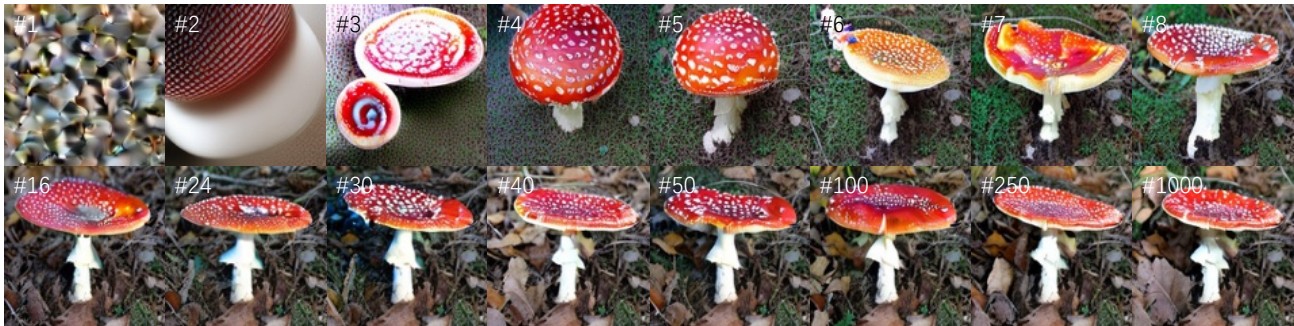

Figure 9. **Samples generated with different numbers of denoising steps.** With only 2 steps, the model already captures the coarse semantics; with 3 steps, salient and recognizable structures emerge. At 5 steps, the full image is largely formed. As the number of denoising steps increases up to 1000, fine-grained background details (*e.g.*, dried leaves) become progressively sharper and more distinguishable.

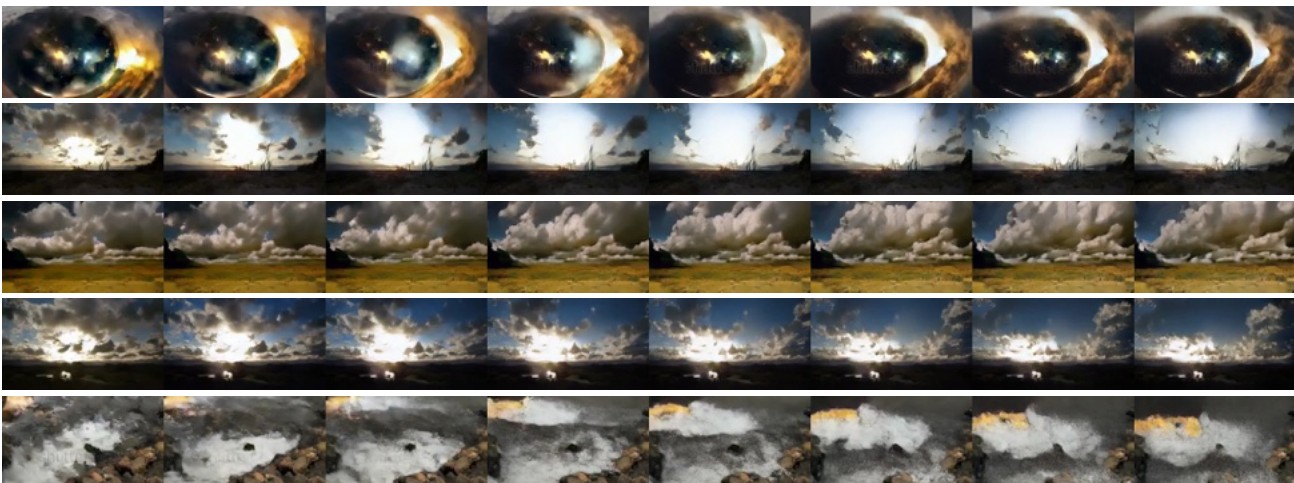

Figure 10. **Video generation with SAR.** We train SAR-L for autoregressive video synthesis via next-frame prediction.

this improvement to the fact that $\gamma$-Noise effectively suppresses excessive local similarity in continuous feature representations, encouraging more diverse and informative features under spatial causality and thereby improving both optimization stability and final generation quality. Please refer to App. 3.4 for further discussion of $\gamma$-Noise.

**Position embedding.** Many autoregressive image generation methods adopt 1D RoPE or learnable positional embeddings; while 2D variants better match image geometry, 1D RoPE can still be effective in practice. We use MM-RoPE, which can be viewed as a fine-grained 2D RoPE that interleaves height and width coordinates in the last dimension. In implementation, we scale the coordinates by 16, effectively aligning them with pixel-space magnitudes, which empirically improves optimization behavior compared to purely learnable position embeddings. Table 5 supports this choice: under the strong setting with both adaLN-Zero and $\gamma$-Noise enabled, replacing MM-RoPE with a learnable positional embedding degrades FID-50k from 6.54 to 7.26.

Overall, the results in Table 5 highlight the importance of explicit step-conditioning and the benefit of combining it with $\gamma$-Noise and MM-RoPE.

## F. Denoising Steps

We employ Flow Matching as the denoising head. Since Flow Matching itself can be a strong image generator, we want to clarify that the generation capability of SAR primarily comes from the *backbone*, rather than being dominated by the denoising head. As analyzed in Sec. 3.3 and evidenced by Tab. 3, the choice of denoising head has a noticeable impact on final quality. However, this does not imply that the denoising head *determines* the model's generative power. In our framework, the denoising head mainly serves as a lightweight renderer that maps the backbone feature $z$ to local image content, while the global structure and compositional relationships of the image are governed by the backbone output $z$.

This interpretation is consistent with the denoising head's

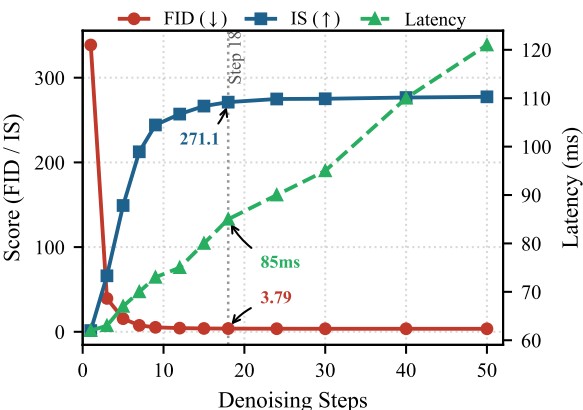

*Figure 11.* **Generation quality vs. latency and denoising steps.** We plot FID50k and IS as a function of the number of denoising steps along with the corresponding end-to-end latency. We find that $\sim$10 steps nearly matches the quality obtained with 50 steps, while keeping latency below 80 ms.

limited capacity: it is typically a simple MLP with fewer than 100M parameters. We further support this claim by varying the number of denoising steps at inference. Figure 9 shows that with roughly 5 iterative steps, the model already produces visually convincing samples. Figure 11 plots FID versus denoising steps and indicates that around 20 steps are sufficient to nearly reach the final performance. Notably, standard Flow Matching models often require about 50 steps to achieve high-quality samples without distillation. Achieving strong results with substantially fewer steps suggests that the backbone representation $z$ carries the essential generative signal, and the denoising head primarily decodes it rather than providing the core modeling capacity.

## G. CFG Curve

As discussed in Sec. 3.2, autoregressive models often exhibit a strong dependence on classifier-free guidance (CFG). To quickly identify effective guidance strengths, we perform a grid search over the CFG parameters. In practice, we first search for the optimal value of $x$-**CFG**, denoted by $\omega_x$, and then fix $\omega_x$ while sweeping $\omega_z$ for the paired $x$-**CFG**+$z$-**CFG** setting. For the standard CFG baseline (a single guidance scale $\omega$), we conduct an independent sweep over $\omega$. We do not tune $z$-**CFG** in isolation; instead, we always combine it with $x$-**CFG**, since extensive experiments show that using $z$-**CFG** alone is rarely optimal. The results reported in Tab. 4 are selected based on empirical performance, choosing the better option between standard **CFG** and the combined $x$-**CFG**+$z$-**CFG**.

Figure 13 shows an example record of our parameter search. Across different guidance strategies, we observe a clear optimum in terms of guidance strength. Once a near-optimal

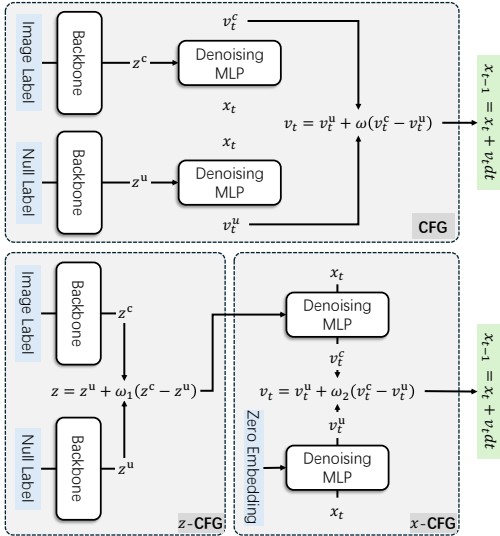

*Figure 12.* **Factorizing standard classifier-free guidance.** Under $v$-prediction, we decompose conventional CFG into two complementary forms: $z$-**CFG**, which applies guidance by interpolating backbone features, and $x$-**CFG**, which applies guidance at the head output by constructing an unconditional branch with $z = 0$.

guidance configuration is found, the impact of denoising steps or the choice of ODE solver on the final quality becomes marginal, and we therefore do not perform additional exhaustive searches.

## H. Implementation Details

Table 6 summarizes the detailed configurations, training hyperparameters, and sampling settings for different models. Based on extensive empirical studies, we draw the following observations.

**Denoising Head Capacity.** We observe that increasing the capacity of the denoising head—*e.g.*, using deeper architectures or larger hidden sizes—generally improves the final performance. However, in the VAE latent space, a denoising head with around 40M parameters is already sufficient. Further increasing the model size brings marginal performance gains while significantly increasing latency. In contrast, for the raw-pixel space, a substantially larger hidden size (*e.g.*, 2048) is necessary. This is because each token in the raw-pixel space has a high-dimensional input feature (up to 768 dimensions), making the denoising task considerably more challenging. Consequently, a larger hidden size is required to effectively model such high-dimensional inputs.

**Prediction Parameterization.** We adopt *x-prediction* for models operating in the raw-pixel space, following prior work (Li & He, 2025). Empirically, x-prediction consistently yields better performance in this setting. Nevertheless, we emphasize that even under *v-prediction*, SAR re-

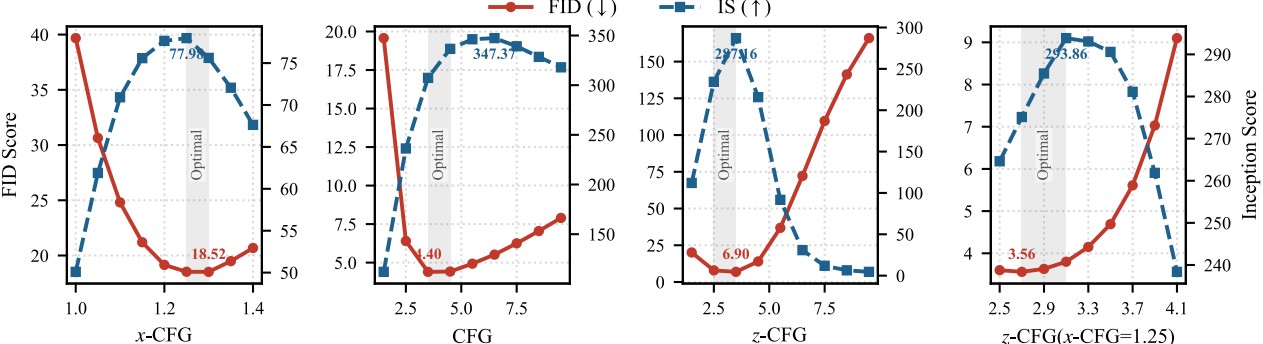

*Figure 13.* **Performance sensitivity to guidance weights.** We report FID-50k and Inception Score (IS) as functions of the guidance scales $\omega$, $\omega_x$, and $\omega_z$ for **CFG**, $x$-**CFG**, and $z$-**CFG**. The resulting curves indicate distinct optimal operating points for each guidance mechanism. All results are obtained using SAR-L trained for 200 epochs (250k iterations), with a flow-matching denoising head run for 30 steps.

mains highly effective in the raw-pixel space, with only a slight performance degradation compared to x-prediction, and without any training instability. For all other representation spaces, v-prediction not only achieves better final performance but also converges faster than x-prediction. Therefore, we use x-prediction exclusively for the raw-pixel space and v-prediction for all other cases.

**Weight Decay.** Unlike many generative models that typically avoid weight decay, we employ a relatively large weight decay of 0.05. This design choice is motivated by our goal of training a unified model that better supports downstream understanding and generation tasks. Our experiments show that the presence or absence of weight decay has negligible impact on the final performance. As a standard practice, we do not apply weight decay to bias terms or linear layers.

$\gamma$-**Noise.** As illustrated in Fig. 7, setting the $\gamma$-Noise coefficient to 0.5 is already sufficient to achieve strong performance. We also experimented with a larger value of 1.0, corresponding to stronger noise injection. While the model remains trainable under this setting, we do not observe any clear performance improvements. Therefore, we fix $\gamma = 0.5$ in all experiments.

**Time-Step Sampling for the Denoising Head.** During training, we find that biasing the sampling of the denoising head time step $\tau$ toward higher-noise regions can improve performance. However, compared to full flow-matching models, the optimal time-shifting for the denoising head is considerably smaller. In practice, we sample $\tau$ from a log-normal distribution controlled by parameters $\mu$ and $\sigma$, as shown in Fig. 14. Using a more aggressive setting with $\mu = -0.8$ and $\sigma = 0.8$, which emphasizes higher noise levels, surprisingly leads to worse performance than

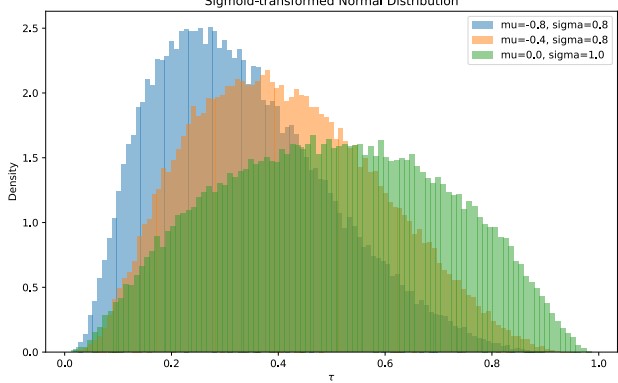

*Figure 14.* **Time shifting.**

uniform sampling. We ultimately find that $\mu = -0.4$ and $\sigma = 0.8$ strike a better balance and yield superior results.

**ODE Solvers for Sampling.** We experiment with both Euler (first-order) and Heun (second-order) ODE solvers during sampling. The resulting evaluation metrics are nearly identical, and in some cases, Euler even slightly outperforms Heun. Considering the reduced computational cost and lower latency, we adopt the more efficient Euler solver by default.

**Classifier-Free Guidance Variants.** In addition to the proposed $x$-**CFG** and $z$-**CFG**, we find that applying a CFG interval significantly improves sampling stability. Consequently, this technique is enabled by default. Moreover, even when $p_2$ is specified, we observe that using standard CFG sampling—without explicitly applying $x$-**CFG**—still leads to noticeable performance improvements. We attribute this behavior to the practice of randomly setting $z$ to a zero embedding, which effectively enhances the training of the

Table 6. **Experimental configurations.**

| Setting | SAR-B | SAR-L | SAR-H | SAR-G |
|---|---|---|---|---|
| **Architecture** | | | | |
| Depth | 12 | 28 | 32 | 40 |
| Hidden size | 768 | 1152 | 1280 | 1664 |
| Heads | 12 | 18 | 16 | 16 |
| Dropout | 0 | 0 | 0.2 | 0.2 |
| Flow head depth | | | 6 | |
| Flow head dims | | Same as hidden size; 2048 for raw-pixel inputs | | |
| MM RoPE scale | | 16, 16 | | |
| **Training** | | | | |
| Epochs | | 200 for ablation; no more than 600 for final | | |
| Warmup epochs | | 5 | | |
| Optimizer | | AdamW, $\beta_1$=0.9, $\beta_2$=0.95, $\epsilon$=$1 \times 10^{-8}$ | | |
| Batch size | | 1024 | | |
| LR | | $1 \times 10^{-4}$ | | |
| LR schedule | | Constant | | |
| Weight decay | | 0.05 | | |
| EMA decay | | 0.9996 | | |
| Denoising time sampler | | $\text{logit}(t) \sim \mathcal{N}(\mu, \sigma^2)$, $\mu = -0.4$, $\sigma$=0.8 | | |
| $p_1$ | | 0.1 | | |
| $p_2$ | | 0.1 | | |
| $\gamma$-Noise | | 0.5 | | |
| **Sampling** | | | | |
| ODE solver | | Euler | | |
| ODE steps | | 30 for ablation; 50 for final | | |
| Time steps | | Linear from 0.0 to 1.0 | | |
| CFG sweep range | | 1.5 to 7.5 with step 1.0 (if used) | | |
| $x$-CFG sweep range | | 1.0 to 1.4 with step 0.05 (if used) | | |
| $z$-CFG sweep range | | 2.5 to 5.5 with step 0.5 (if used) | | |
| CFG interval | | [0.1, 1.0] (if used) | | |

unconditional branch.

## I. Training Curve

Figure 15 illustrates the evolution of the gradient norm and the flow-matching loss during the training of SAR-L. In general, the model converges rapidly. After approximately 5k iterations, it is already able to generate images that are largely free of noise. By around 50k iterations (corresponding to roughly 40 epochs), the FID score typically drops to around 10, after which further improvements become gradual.

In contrast, when the model design is suboptimal—such as using an inappropriate dependency injection function $h(\cdot)$ or an unsuitable grouping strategy—the training dynamics degrade noticeably. In such cases, convergence becomes significantly slower, and even after 100k iterations, the FID may remain above 100. These observations indicate that the convergence speed serves as a practical diagnostic signal for assessing whether the model configuration is well-posed.

We also observe that the absolute scale of the flow-matching loss varies substantially depending on the choice of VAE as well as whether v-prediction or x-prediction is used. Nevertheless, as long as the loss exhibits stable convergence, the training process is generally well-behaved. As training pro-

gresses, the gradient norm gradually stabilizes and typically converges to a value around 0.05.

## J. Additional Visualizations

Figs. 17, 18, 19, and 20 present randomly generated samples produced by SAR-L under different configurations, corresponding to the settings discussed in Sec. 2.5. We note that these samples appear noticeably inferior in visual quality and may contain substantial artifacts or structural errors. This is expected, as the corresponding models are not fully trained and are intended primarily for controlled empirical analysis rather than final-quality generation.

Fig. 21 shows additional uncurated samples generated by SAR-L. With sufficient training, the generation quality improves significantly, yielding substantially more coherent and visually appealing results.

## K. Limitations and Discussion

While *self-token prediction* extends next-token prediction and enables efficient generation of a large number of tokens under strict *spatial causality*, several limitations remain. First, for emerging modalities, we still lack principled design guidelines—including how to perform effective token grouping and how to construct an appropriate dependency injection function $h(\cdot)$. In practice, these choices are highly modality-dependent and currently require extensive empirical validation.

Second, our experiments are primarily confined to relatively simple image generation settings. How well the proposed paradigm generalizes to more challenging scenarios, such as image editing, or to truly multimodal fusion involving audio and video, remains unclear and calls for systematic study.

Third, our motivation for self-token prediction is rooted in the hypothesis that preserving spatial causality is a key ingredient behind the success of large language models. Consequently, we focus on improving next-token-style training and inference *without* relaxing causal constraints. In contrast, alternatives such as masked prediction or next-scale prediction often trade off spatial causality for convenience or parallelism. Although we achieve strong results on ImageNet generation, whether strict spatial causality is indeed fundamental to later-stage capability emergence remains an open question.

Despite these limitations, we believe self-token prediction offers a promising new direction for constructing unified multimodal models: it maintains a causal, autoregressive learning signal while expanding the design space for efficient token generation.

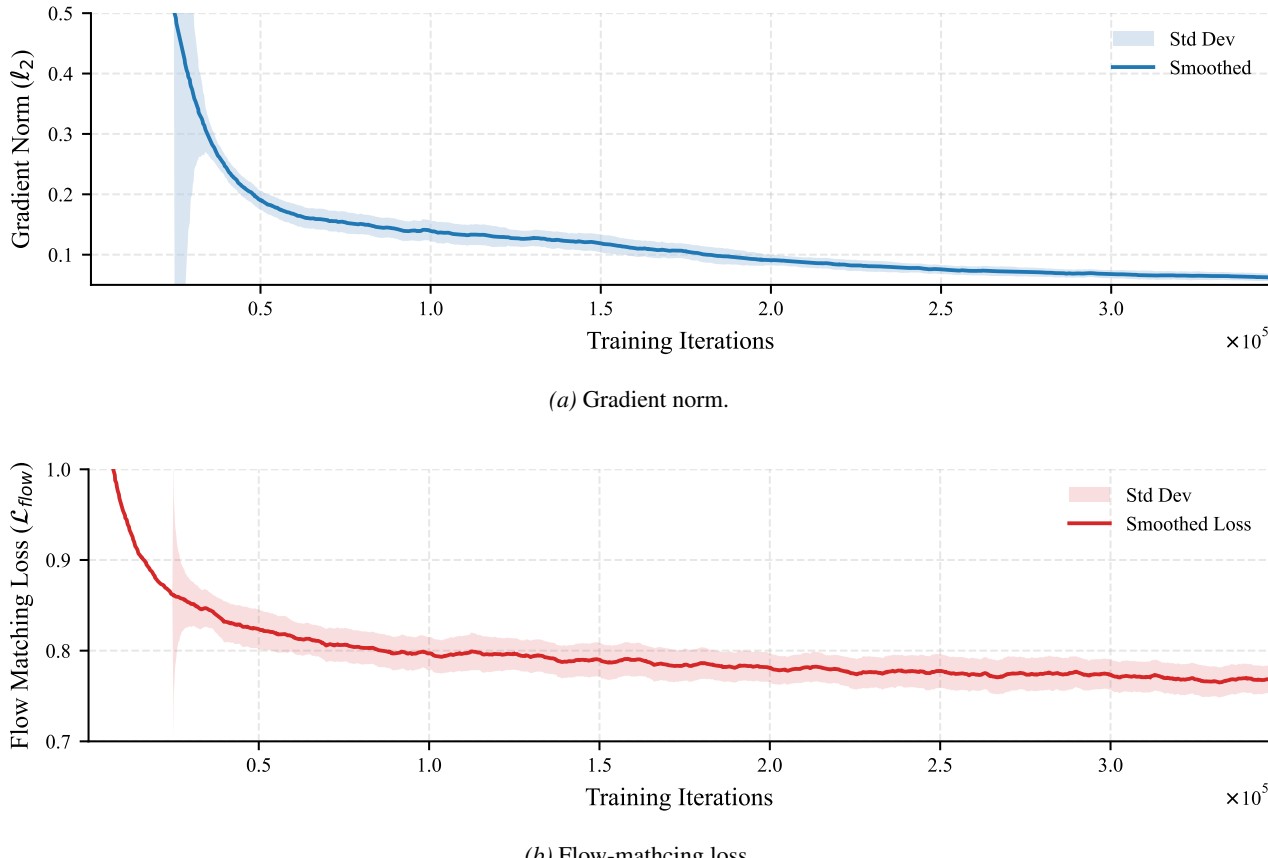

*(a)* Gradient norm.

*(b)* Flow-mathcing loss.

*Figure 15.* **Training curves of SAR-L. (a)** Gradient norm steadily decreases during training. **(b)** Flow-matching loss remains stable. Self-token-prediction-based autoregressive training mitigates the loss spikes often seen in LLM training.

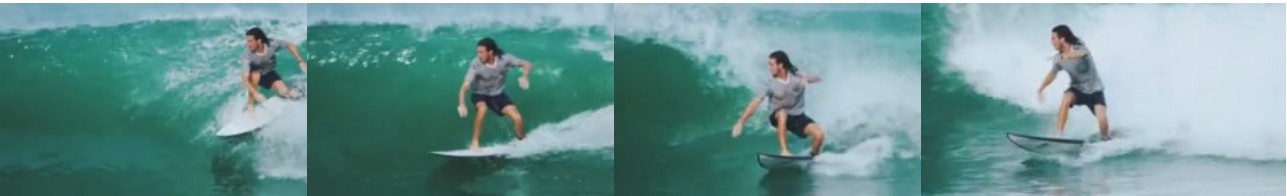

A man is surfing on a white surfboard with a black stripe on the bottom, riding a large, green wave. He is wearing a striped shirt and black shorts. The man maintains his balance by adjusting his stance and using his arms for stability. The wave behind him is large and crashing, creating a dynamic and energetic background. The man continues to surf smoothly, making slight turns and adjustments to his position on the board. The water is a vibrant green, indicating a clear and sunny day. The man skillfully maneuvers through the wave, showcasing his surfing skills. The man maintains his balance and control over the surfboard as he rides the wave.

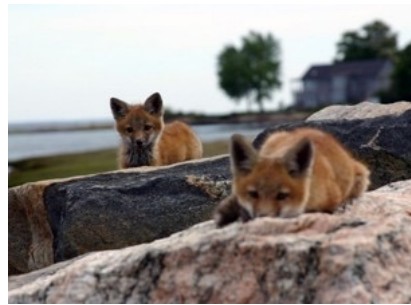

**Two foxes** *resting* on **large rocks**. The foreground features a fox *lying down*, with its head resting on the rock. The second fox is *sitting upright* on a rock in the midground, *looking directly* at the camera. Both foxes have a reddish-brown fur, with darker shades around their eyes and ears.
In the background, there is **a body of water**, possibly a lake or a calm sea. Beyond the water, there is a blurred landscape that includes **a few trees** and **a house or building**. The sky is overcast, with a soft, diffused light that gives the scene a serene and peaceful atmosphere.

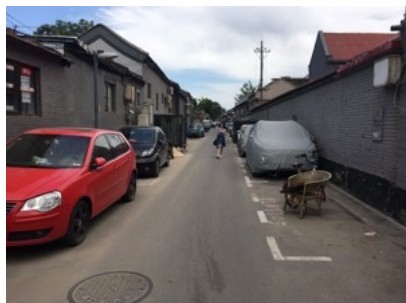

A street scene in what appears to be a residential area. The photograph is taken from a slightly elevated angle, giving a wide-angle view of the street.
On the **left side** of the street, there **is a red car** parked close to the curb. **Behind** the red car, there are **several other vehicles** parked along the street. The buildings on this side of the street are made of brick and have a traditional architectural style. Some of the buildings have signs and windows, indicating that they are likely residential homes.
On the **right side** of the street, there is **a large, gray car** parked along the curb. In front of this car, **a person** is walking down the street, heading towards the camera. The person is wearing casual clothing. There appears to be **a small, wooden chair** placed on the sidewalk. The chair is empty and seems to be a makeshift object. The street itself is paved and appears to be relatively narrow. There are **white parking lines** on the road, indicating designated parking spaces. The street is lined with buildings on both sides, creating a somewhat enclosed feel.
In the background, there are more buildings, some with **red roofs.** The sky is partly cloudy, suggesting it might be a day with mixed weather conditions.

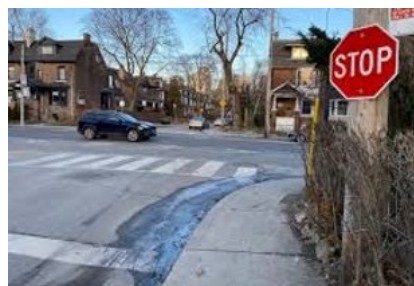

A quiet residential street with **a stop sign** on the right side. a car is driving down the street, which is lined with houses and trees. the sky is clear, suggesting a calm, sunny day.

*Figure 16.* **Visual understanding with SAR.** We train SAR-L for image and video understanding.

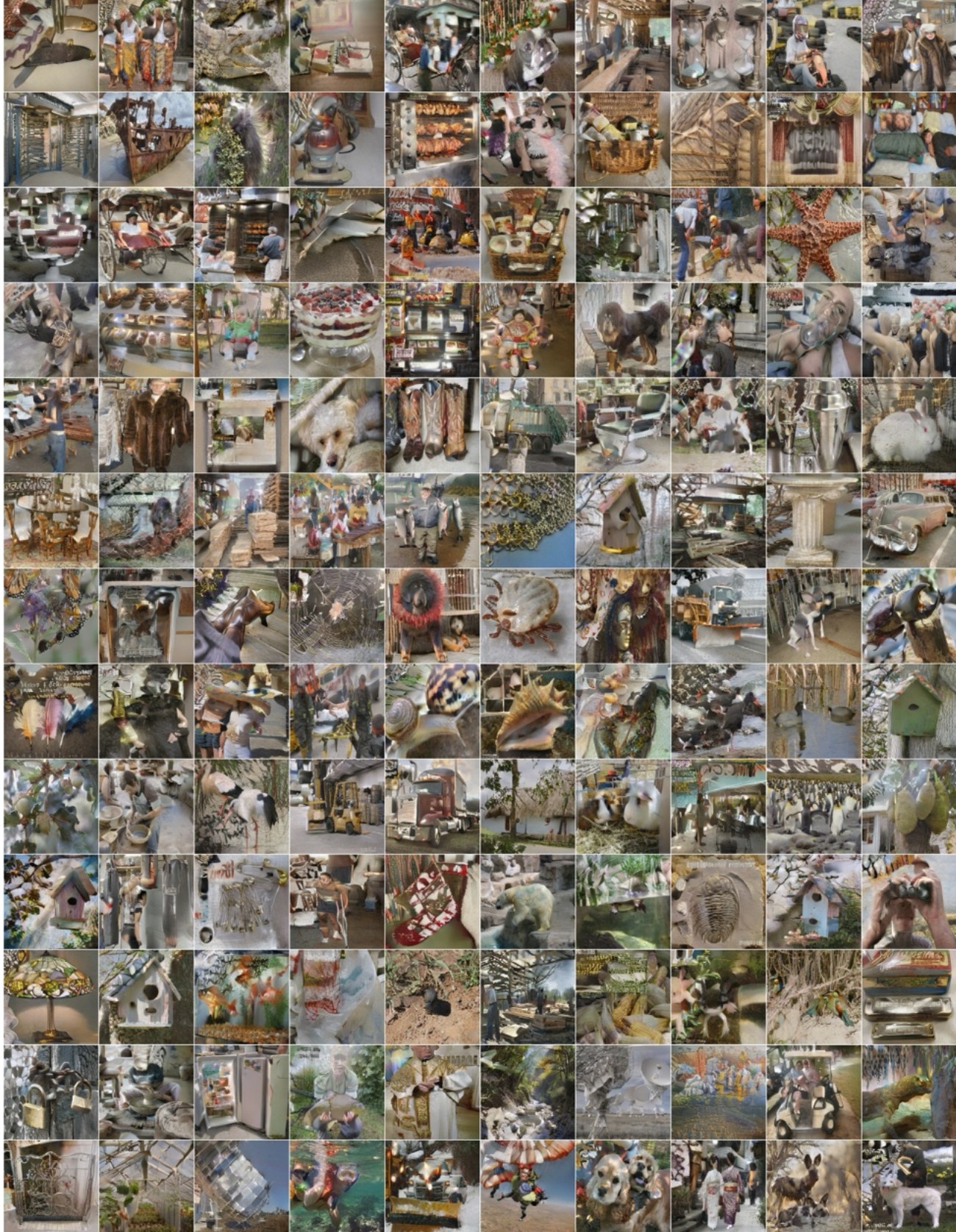

*Figure 17.* **Uncurated samples from SAR-L on ImageNet (256×256).** Images are generated in the image latent space(Rombach et al., 2022) using a DDPM denoising head (Nichol & Dhariwal, 2021). The model is trained for 250k iterations (200 epochs).

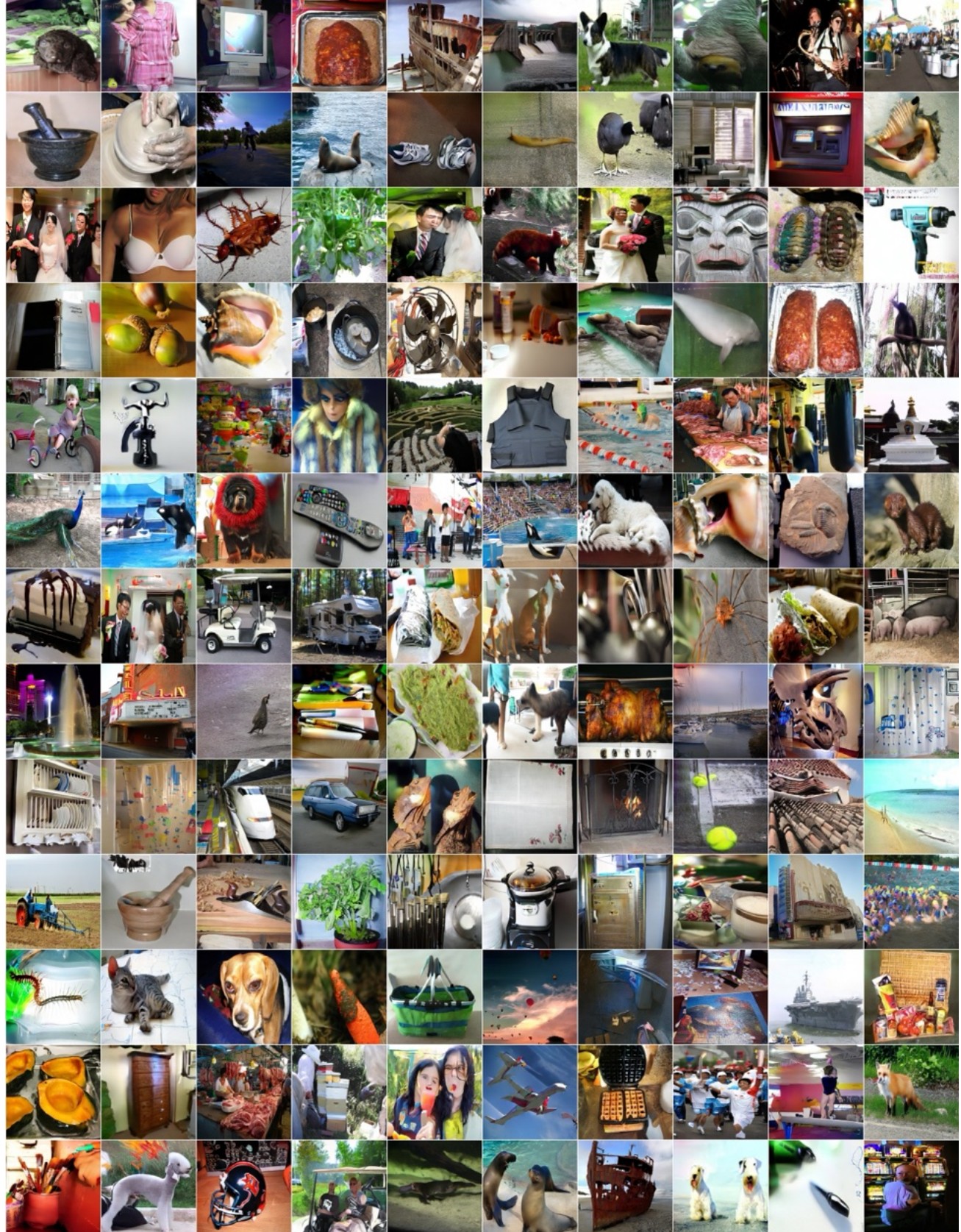

*Figure 18.* **Uncurated samples from SAR-L on ImageNet (256×256).** Images are generated in the image-video latent space (Wan et al., 2025) using a flow-matching denoising head (Lipman et al., 2022). The model is trained for 250k iterations (200 epochs).

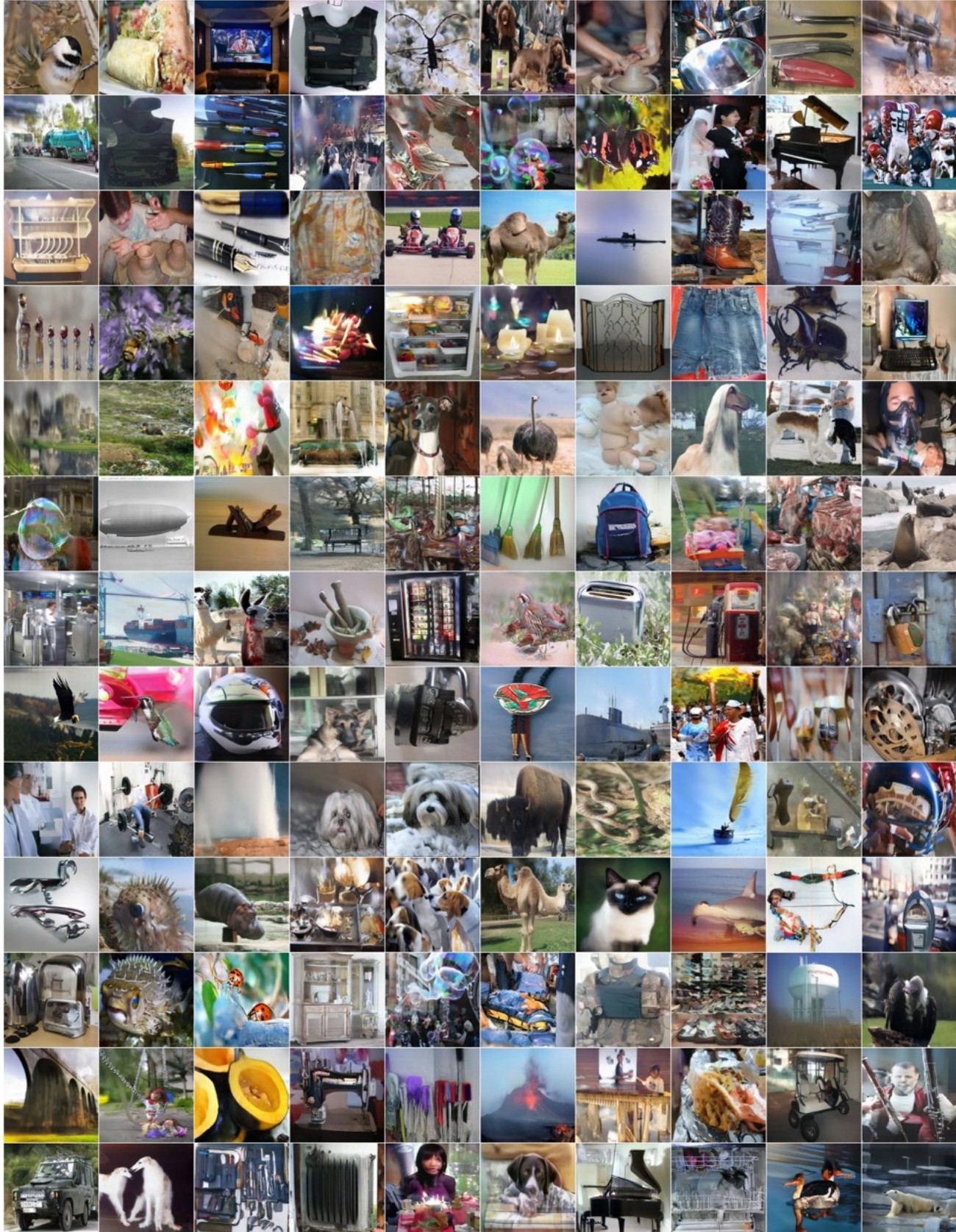

*Figure 19.* **Uncurated samples from SAR-L on ImageNet (256×256).** Images are generated in the raw-pixel space (Li & He, 2025) using a flow-matching denoising head (Lipman et al., 2022). The model is trained for 250k iterations (200 epochs).

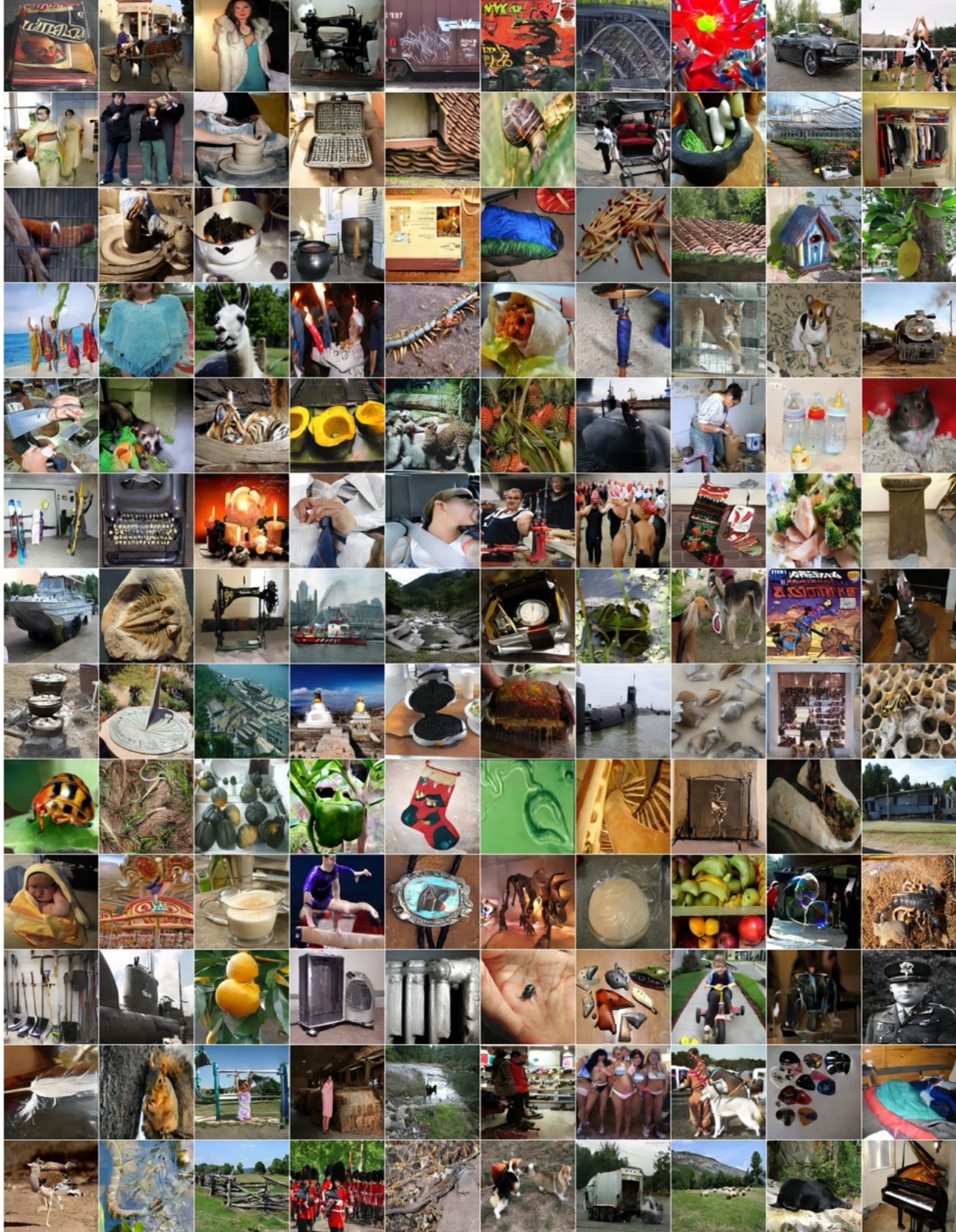

*Figure 20.* **Uncurated samples from SAR-L on ImageNet (256×256).** Images are generated in the image latent space (Rombach et al., 2022) using a flow-matching denoising head (Lipman et al., 2022). The model is trained for 250k iterations (200 epochs).

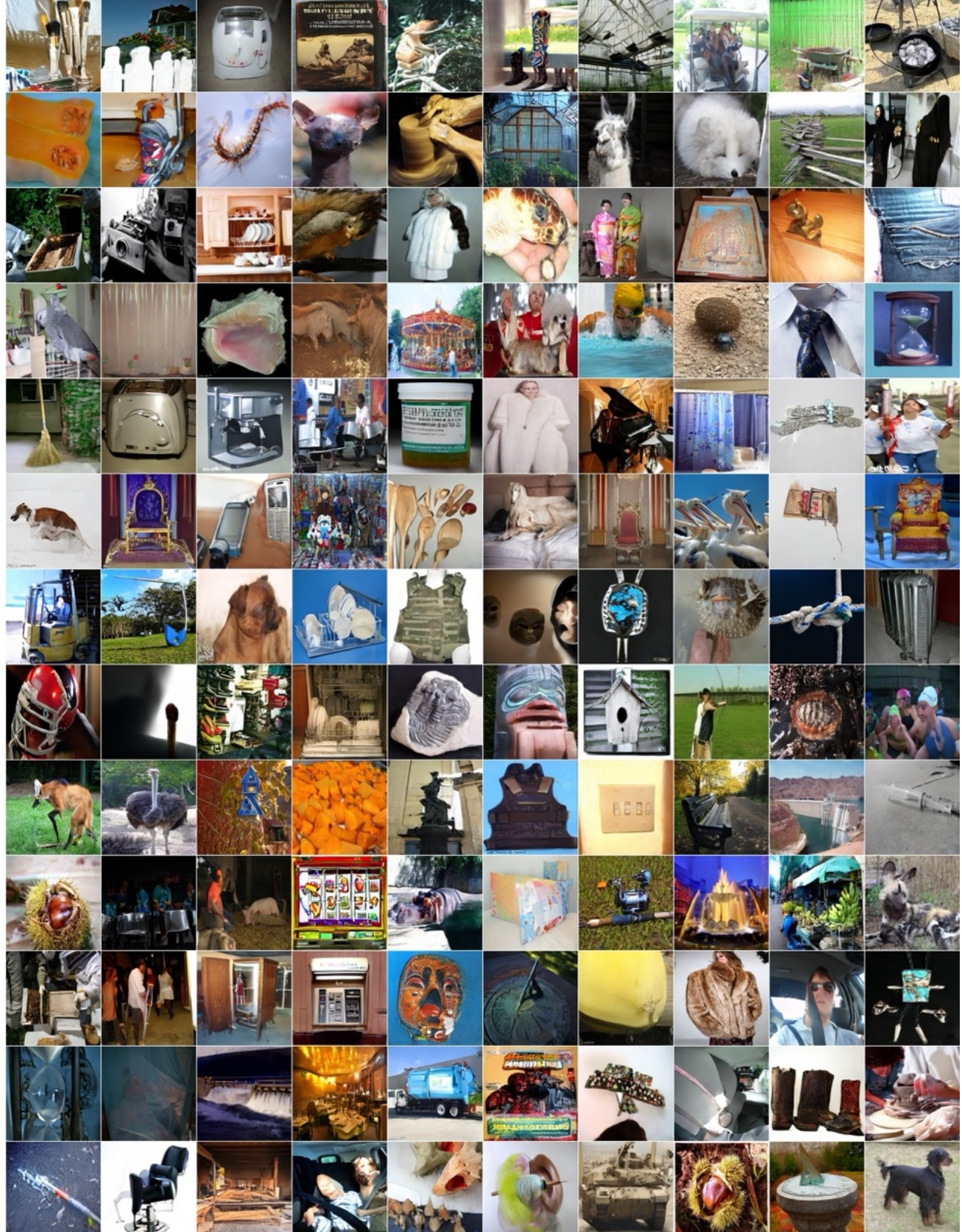

*Figure 21.* **Uncurated samples from SAR-L on ImageNet (256×256) with sufficient training.**

