# OpenReview forum: "Autoregression with Self-Token Prediction"
_ICML.cc/2026/Conference — ICML 2026 regular_

### Official Review · Reviewer_hPer · 2026-03-13

**Soundness:** 2
**Presentation:** 2
**Significance:** 2
**Originality:** 2
**Overall Recommendation:** 3
**Confidence:** 4

**Summary:**

This paper proposes self-token prediction, which relaxes the next-token constraint in autoregressive generation by predicting a flexible number of tokens per step while preserving strict spatial causality through group-wise dependency design. Based on this idea, the paper introduces AGARIC, a spatially causal image generator, and shows strong ImageNet 256×256 results with substantially fewer autoregressive steps. The paper further argues that this framework is a promising direction toward efficient and unified multimodal autoregressive modeling.

**Compliance With Llm Reviewing Policy:**

Affirmed.

**Key Questions For Authors:**

1. In Abstract / Sec. 1, the paper claims that the method is “a key step toward unified and efficient multimodal autoregressive modeling,” and later says it is “a highly scalable solution for next-generation multimodal foundation models.” However, the main validation is on class-conditional ImageNet image generation. Could the authors moderate this claim or add stronger multimodal evidence? Relevant text: “Our findings point to self-token prediction as a key step toward unified and efficient multimodal autoregressive modeling.” and “establishing it as a highly scalable solution for next-generation multimodal foundation models.”

2. In Sec. 2.3, the paper states that training “roughly doubles the number of processed tokens during training.” In Sec. 2.4, the method also requires cache reconciliation and recomputation during inference. Could the authors provide a fuller training/inference cost analysis, such as wall-clock training time, memory, FLOPs, and end-to-end throughput? Relevant text: “roughly doubles the number of processed tokens during training” and “we discard the KV entries associated with G˜t and recompute them.”

3. In Sec. 4.2, the paper argues that the computational overhead is “significantly lower than non-causal approaches,” and in the abstract it claims “markedly faster inference.” But the paper does not show a unified wall-clock comparison against strong non-causal baselines such as DiT / MAR under the same hardware and implementation conditions. Could the authors provide direct runtime comparisons? Relevant text: “AGARIC delivers markedly faster inference” and “the computational overhead is significantly lower than non-causal approaches.”

4. In Sec. 3.1, the method fixes a specific spatial design: “We instantiate spatial causality via a Manhattan-distance grouping,” and in Eq. (7) uses a hand-crafted local aggregation rule. How sensitive are the results to these choices, and is there any more principled guidance for choosing grouping and dependency constructors for other modalities such as video or multimodal sequences? Relevant text: “We instantiate spatial causality via a Manhattan-distance grouping” and “the backbone input is constructed by aggregating causally available neighbors.”

5. In Sec. 4 / Table 4, the main benchmark is ImageNet-1K 256×256. Could the authors discuss whether the conclusions still hold in broader settings such as higher resolution, text-to-image, or real video generation benchmarks? Relevant text: “We train on ImageNet-1K with 256×256 resolution” and “Comparison with state-of-the-art models on ImageNet-1K at 256 × 256.”

**Limitations:**

yes

**Strengths And Weaknesses:**

# Strength
The paper studies an important problem: reducing the inefficiency of next-token autoregression for high-dimensional visual generation. The formulation is fairly clear, especially the decomposition into token grouping, dependency constructor, and causal masking. The empirical results are strong for strictly spatially causal image generation, and the ablations are useful in showing that dependency design, not only fewer steps, is critical to performance. In particular, AGARIC-H reaches competitive FID on ImageNet while using only 31 autoregressive steps.

# Weaknesses
Although the abstract and introduction claims the method as a step toward “unified and efficient multimodal autoregressive modeling,” the main experiments are almost entirely limited to class-conditional ImageNet image generation. The efficiency claims are also not fully convincing, because the paper mainly reports step counts and limited latency numbers, without a more systematic end-to-end comparison against strong non-causal baselines under a unified runtime setting. In addition, the method introduces nontrivial training and inference overheads such as duplicated streams during training and cache reconciliation during sampling, but the paper does not provide a sufficiently complete cost analysis. Some key design choices also remain fairly heuristic, such as the Manhattan grouping and the specific dependency constructor.

---

> ### Author Rebuttal · Authors · 2026-03-30
>
> ## 1. Overclaim on Multimodal Generalization
>
> Our work is indeed motivated by building a **unified multimodal autoregressive framework based on self-token prediction**. However, due to space constraints, we focused the main paper on **ImageNet experiments** to clearly validate the core idea of self-token prediction and maintain a coherent narrative.
>
> That said, we do include additional multimodal evidence in the appendix:
>
> * **Figure 10** presents qualitative results on **video generation**.
> * **Figure 16** demonstrates **image and video understanding capabilities**.
>
> We agree that some of our wording in the main text may have overstated the multimodal aspect relative to the presented evidence. In the final version, we will **revise the phrasing to better balance ambition and evidence**, emphasizing the contribution of **self-token prediction itself**, while moderating claims about unified multimodal modeling.
>
> ---
>
> ## 2. Incomplete Efficiency & Cost Analysis
>
> For additional details, we refer to our responses to other reviewers (e.g., Hkkz Q5 and eDu6 Q3), where we discuss training and inference overhead, including the dual-stream design.
>
> In the paper:
>
> * **Table 1 and Table 2** report **measured sampling latency**, reflecting real end-to-end runtime.
> * **Table 4** compares **sampling steps** across methods as a proxy for autoregressive efficiency.
>
> We intentionally emphasize **wall-clock latency** as a primary metric. While some autoregressive methods report very low FLOPs (e.g., 1/10 or even 1/100 of diffusion models), their **actual latency can still be significantly higher** due to sequential dependencies. We believe latency better reflects practical system performance.
>
> To improve transparency, we have also released detailed training logs (including training time and related metrics) at:
>
> * [https://anonymous.4open.science/r/ICML26-D03D/](https://anonymous.4open.science/r/ICML26-D03D/)
>
> In the final version, we will aim to **expand the discussion on training/inference cost**, including clearer reporting of computational overhead.
>
> ---
>
> ## 3. Lack of Fair Runtime Comparison with Baselines
>
> In **Table 4**, we report the **number of sampling steps** required by different methods:
>
> * For autoregressive models: number of decoding steps
> * For diffusion models: number of denoising steps
>
> Since each step typically incurs a roughly similar computational cost within the same model family, step count serves as a **useful and intuitive proxy for latency**.
>
> That said, we agree that **direct wall-clock comparisons under unified hardware and implementation settings** would further strengthen the evaluation. We will clarify this point in the paper and, where possible, include more explicit runtime comparisons in the final version.
>
> ---
>
> ## 4. Heuristic Design Choices (Lack of Generality)
>
> Additional discussions on **Manhattan-distance grouping** and the **dependency constructor** can be found in our responses to reviewer Hkkz (Q1). We also provide further empirical evidence in **Appendix D (More Empirical Studies)**, including results on other modalities such as video.
>
> Importantly, our findings suggest that:
>
> * The specific **token grouping strategy is not the primary factor** determining performance.
> * Instead, the key lies in whether the model can **effectively inject dependencies under causal constraints**, enabling it to learn meaningful structure.
>
> In the paper, we adopt a **local aggregation–based dependency constructor** for its simplicity and interpretability. However, this is not a strict limitation. The framework can be extended to more general designs. For example:
>
> * One can use a **Flamingo-style conditioning mechanism**, where tokens from the previous group are treated as context and incorporated via in-context conditioning.
>
> This approach:
>
> * Removes the need for hand-crafted aggregation rules
> * Potentially eliminates the dual-stream design
>
> We have validated this alternative in text-to-image experiments:
>
> - https://anonymous.4open.science/r/ICML26-D03D/avg-vs-flamingo-loss-curve.png
>
> It shows **improved convergence behavior**.
> We also observe improved results in discrete image generation, achieving:
> * **FID 2.13 vs. 2.34** (compared to 1.4B LlamaGen-XXL on ImageNet)
> * While requiring only **31 sampling steps vs. 256**
> We will clarify in the final version that our current design is **one instantiation**, and that the framework itself is **more general and extensible**.
> ---
>
> ## 5. Limited Experimental Scope
>
> In **Appendix D (More Empirical Studies)**, we provide additional qualitative results on other modalities. However, we acknowledge that a more comprehensive evaluation (e.g., quantitative benchmarks on video or text-to-image tasks) is beyond the current scope due to time and resource constraints. More importantly, we will **revise the paper to better align the claims with the presented evidence**, focusing the main narrative on validating **self-token prediction** itself.

---

> > ### Author Rebuttal · Reviewer_hPer · 2026-04-04
> >
> > I appreciate the rebuttal, while the argument around efficiency is not well taken. Both latency and FLOPs matter to justify the superiority of a method, and typically autoregressive models decode at large batch sizes while diffusion models denoise a single sample. It would be interesting to see the trade-off, i.e., how much more FLOPs is it using in return for a lower latency.

---

> > > ### Author Response · Authors · 2026-04-06
> > >
> > > **Response to efficiency trade-off (latency vs FLOPs):**
> > >
> > > We thank the reviewer for raising this important point. We agree that both latency and FLOPs are necessary to fairly assess efficiency. Below we provide a simple analytical comparison to clarify the trade-off.
> > >
> > > To make the comparison concrete and fair, we consider three paradigms under a unified setting:
> > >
> > > * Same model size
> > > * Same target image with ($N$) tokens
> > > * Let ($\tau$) denote the compute cost of processing one token once
> > > * Let ($s$) denote the latency of one forward pass (assuming sufficient GPU parallelism)
> > >
> > > ---
> > >
> > > ### Theoretical comparison
> > >
> > > | Method                   | #Steps | Total FLOPs            | Latency      |
> > > | ------------------------ | ------ | ---------------------- | ------------ |
> > > | Diffusion                | ($T$)    | ($T \cdot N \cdot \tau$) | ($T \cdot s$)  |
> > > | Next-token AR            | ($N$)    | ($N \cdot \tau$)         | ($N \cdot s$)  |
> > > | **Self-token AR (ours)** | ($T'$)   | ($N \cdot \tau$)         | ($T' \cdot s$) |
> > >
> > > ---
> > >
> > > ### Key observations
> > >
> > > 1. **FLOPs (total compute):**
> > >
> > >    * Diffusion requires **($T\times$)** more compute, since each token is processed at every denoising step.
> > >    * Both autoregressive methods (next-token and ours) process each token **only once**, resulting in **($N \cdot \tau$)** total compute.
> > >    * Therefore, **our method does not increase total FLOPs compared to standard AR**, and is significantly more compute-efficient than diffusion.
> > >
> > > 2. **Latency:**
> > >
> > >    * Next-token AR suffers from **strict sequential dependency**, leading to ($N$) steps.
> > >    * Diffusion reduces steps (($T \approx 30\sim50$)), but still requires multiple full passes.
> > >    * Our method reduces the number of autoregressive steps to ($T'$) (e.g., ($\sim 30$)), while still maintaining full-token coverage per step.
> > >
> > > 3. **Trade-off:**
> > >
> > >    * Compared to diffusion:
> > >
> > >      * Similar latency scale (($T' \approx T$))
> > >      * **Significantly lower FLOPs** (($\times T$) reduction)
> > >    * Compared to next-token AR:
> > >
> > >      * Same FLOPs
> > >      * **Much lower latency** (($T' \ll N$))
> > >
> > > ---
> > >
> > > ### Intuition
> > >
> > > The key distinction is:
> > >
> > > * Diffusion: **repeatedly processes all tokens → high FLOPs**
> > > * Next-token AR: **processes tokens once but sequentially → high latency**
> > > * **Ours: processes tokens once AND in parallel groups → low FLOPs + low latency**
> > >
> > > ---
> > >
> > > ### Clarification on batching concern
> > >
> > > We agree that batching and hardware utilization can affect practical runtime. Our analysis assumes sufficient parallelism (as commonly used in modern GPU inference). Under this setting, latency is primarily determined by the number of sequential steps, which our method explicitly reduces.
> > >
> > > We will clarify this assumption and add more discussion on FLOPs vs. latency trade-offs in the final version.

---

### Official Review · Reviewer_eDu6 · 2026-03-13

**Soundness:** 3
**Presentation:** 3
**Significance:** 3
**Originality:** 2
**Overall Recommendation:** 4
**Confidence:** 3

**Summary:**

This paper revisits autoregressive modeling for high-dimensional visual data and argues that autoregression need not be restricted to next-token prediction. Instead, it proposes self-token prediction, where the model predicts a group of tokens per autoregressive step while preserving strict causal structure through spatial grouping and masking. Based on this idea, the paper introduces AGARIC, a causal image generation framework with group-conditioned AdaLN, stochastic token noise injection, and modified classifier-free guidance. Experiments show that the method is faster than conventional autoregressive baselines while remaining competitive with strong non-causal generators, and the paper also explores extensions to video generation and visual understanding.

**Compliance With Llm Reviewing Policy:**

Affirmed.

**Final Justification:**

Overall, this paper presents a novel and meaningful revisit of autoregressive modeling for high-dimensional visual data. The idea of self-token prediction is well motivated, technically interesting, and supported by strong empirical results on image generation, with promising extensions to video and visual understanding. The authors’ rebuttal has effectively addressed my concerns. I therefore maintain my Weak Accept recommendation and increase my confidence in this assessment.

**Key Questions For Authors:**

1. How much of the gain comes from self-token prediction itself versus the additional training modifications?
In particular, can the authors better isolate the contributions of grouping, token noise injection, and the modified CFG design?

2. How sensitive is the method to the specific token grouping strategy?
Since the central idea depends on grouping structure, it would be helpful to understand whether performance is robust across alternative group constructions.

**Limitations:**

Yes.

**Strengths And Weaknesses:**

Strengths:
1. The paper generalizes traditional autoregression into a more flexible group-based autoregressive framework. It shows how the proposed formulation can degenerate into conventional next-token prediction or next-scale prediction under appropriate constraints.

2. The self-token prediction framework is clearly defined through token grouping, a causal constructor, and a spatially valid attention mask. AGARIC also includes several practical training choices that are well connected to the proposed formulation, including group-index-conditioned AdaLN and modified CFG.

3. The paper evaluates image generation, discusses transfer across different representation spaces, and includes exploratory extensions to video generation and visual understanding. This helps support the claim that the proposed formulation may be broadly useful rather than narrowly tuned to one setup.



Weaknesses:

1. The model requires two aligned copies for every group (context and generation streams). This effectively doubles the number of tokens processed, increasing the computational and memory burden during training.

2. Key performance drivers, such as $\gamma$-Noise for suppressing local shortcuts and factorized CFG ($z$-CFG/$x$-CFG) , appear somewhat heuristic and may require extensive modality-specific tuning.

---

> ### Author Rebuttal · Authors · 2026-03-30
>
> ### 1. Attribution of Performance Gains
>
> * **Main concern:** It is unclear how much of the improvement comes from the core idea of *self-token prediction* versus auxiliary techniques.
>
> First, we would like to clarify that beyond empirical performance gains, the primary contribution of *self-token prediction* is conceptual. Our results suggest that the key factor underlying strong autoregressive (AR) performance is the ability to learn **effective spatial causal dependencies**, rather than strictly adhering to a one-token-at-a-time generation paradigm.
>
> Building on this insight, we introduce AGARIC, which enables predicting a *group of tokens per step*. This significantly improves generation efficiency, particularly for high-resolution images and videos, where conventional AR methods are inherently slow.
>
> Importantly, the core of self-token prediction lies in constructing an effective **spatial causality modeling mechanism**, rather than in the specific token grouping strategy. As discussed in Sec. 2.5 (Empirical Study), token grouping mainly determines the number of sampling steps, while the critical component is the **dependency injection function**, which governs how causal structure is imposed across tokens. This mechanism is the main driver of both performance and efficiency.
>
> ---
>
> ### 2. Sensitivity to Token Grouping Strategy
>
> Please refer to our response to Reviewer Hkkz (Question 1) for a more detailed discussion. In brief, we find that the method is **robust to different grouping strategies**, as long as the underlying spatial causality is properly preserved.
>
> ---
>
> ### 3. Computational and Memory Overhead
>
> * **Main concern:** The dual-stream design (context + generation tokens) doubles token processing.
>
> We would like to clarify that the **dual-stream design is not required** by self-token prediction itself. Instead, it is one possible implementation tied to a particular form of the dependency injection function.
>
> For example, in a *next-row prediction* setup, where tokens from the current row are directly used to predict the next row, a dual-stream design is unnecessary. We provide empirical evidence for this in Appendix D (More Empirical Studies).
>
> Furthermore, the dual-stream mechanism is only activated during **generation**. When images are used purely as context (e.g., in visual understanding tasks), no additional computation is introduced (see Sec. 2.3, Efficient Training).
>
> Finally, we emphasize that adopting a dual-stream design is **not inherently tied to achieving strong performance**. We present it as a general and flexible framework for implementing self-token prediction under arbitrary dependency structures. As shown in Appendix D, under certain conditions, the dual-stream design can be completely avoided.
>
> To further demonstrate this, we experimented with an alternative dependency injection strategy: replacing the neighbor-based aggregation with a *Flamingo-style in-context conditioning*. This removes the need for dual streams and leads to **faster convergence** in text-to-image tasks. We provide the corresponding loss curve here:
>
> * [https://anonymous.4open.science/r/ICML26-D03D/avg-vs-flamingo-loss-curve.png](https://anonymous.4open.science/r/ICML26-D03D/avg-vs-flamingo-loss-curve.png)
>
> ---
>
> ### 4. Heuristic Design Choices
> - **Token noise injection (to mitigate local shortcuts):**
>   We provide additional analysis and visualizations in Appendix C (*Mitigating Local Shortcut via Noise Injection*). This technique is also supported by subsequent works (e.g., NextStep1.1), suggesting it is not specific to our method but reflects a broader phenomenon in AR modeling.
>
> - **Modified CFG formulation:**
>   We include a detailed analysis in Appendix G (*CFG Curve*), where Figure 13 presents grid search results illustrating the behavior and stability of the proposed formulation.
>
> Overall, both the noise injection and modified CFG strategies have been validated beyond image generation. We observe consistent benefits in other modalities, including **video and audio**, with representative video generation results shown in Figure 10. This suggests that these design choices are not narrowly tuned heuristics, but rather broadly applicable techniques.
>
> ---
>
> ### 5. Clarity on Core Contribution
>
> The central contribution of *self-token prediction* is to generalize traditional **next-token prediction** to **next-group prediction**, enabling autoregressive models to generate multiple tokens per step in a principled and efficient manner.
>
> This significantly improves scalability for high-dimensional data such as high-resolution images and videos, without relying on complex auxiliary generation schemes. At the same time, our formulation **fully preserves the properties of standard autoregression**, including strict causality and likelihood-based training.
>
> We believe this provides a simple yet powerful extension of the AR paradigm, opening up new opportunities for efficient generation across modalities.

---

> > ### Author Rebuttal · Reviewer_eDu6 · 2026-04-04
> >
> > The authors’ rebuttal has addressed most of my concerns, which is consistent with my overall recognition of the value of this work in my original review. Therefore, I would like to maintain my Weak Accept rating.

---

### Official Review · Reviewer_Hkkz · 2026-03-20

**Soundness:** 3
**Presentation:** 3
**Significance:** 3
**Originality:** 3
**Overall Recommendation:** 4
**Confidence:** 4

**Summary:**

This manuscript proposes self-token prediction, which is a novel autoregressive model that generates a flexible number of tokens at each autoregressive step, instead of a single one token. Meanwhile, the authors also introduce AGARIC, which is a spatially causal image generator. Several innovations include a group-wise casual factorization with grouping strategies, an efficient training strategy with duall streams and group-casual attention masks, decomposition of CFG into x-based one and z-based one. Extensive experiments show that proposed innovatoins are solid across multiple representation spaces and superior scaling properties.

**Compliance With Llm Reviewing Policy:**

Affirmed.

**Key Questions For Authors:**

See 'Strengths And Weaknesses'.

**Limitations:**

See 'Strengths And Weaknesses'.

**Strengths And Weaknesses:**

Strengths:

1. This paper is of good novelty by proposing a novel paradigm of self-token prediction, extending original next-token ARMs into group-wise parallel generation, which may be a valuable extension to the existing framework. Together with interesting decomposition of CFG by factorizing original CFG into two formats, offering flexible guidance.

2. The problem formulation is well motivated. The introduction of background effectively shows detailed comparison with next-token and other mechanisms, precisely pointing out the core challenge.

3. The methodological design is consistent with specific problems, forming a complete and logically consistent pipeline.

4. Comprehensive experiments include comparison with SOTAs with dimensions including parameter count, steps, FID, and IS, providing thorough benchmark. Ablation study is comprehensively conducted for example the strong evidence for the effectiveness of CFG decomposition.

Weaknesses:

1. Although experimental results show effectiveness, there is no direct theoretical analysis explaining why Manhattan distance outperforms other spatial groupings.

2. The authors claim that recomputation can be hidden, but they do not provide concrete scheduling strategies or empirical overhead measurements.

3. The claim related to scaling law is somewhat weak. The authors claim that AGARIC exhibits scaling law, but Table 4 shows only three model sizes, which may be encouraged to provide more model sizes.

4. I am curious about what is the core method on text data, and whether this method effective in text with discrete properties.

5. More training costs are encouraged to report by adding absolute training time (GPU-hours) and training FLOPs to help readers evaluate more comprehensively.

---

> ### Author Rebuttal · Authors · 2026-03-30
>
> ### **1. Why does Manhattan distance grouping work better than other spatial grouping strategies?**
>
> We provide additional empirical analysis of different grouping strategies in Appendix D (*More Empirical Studies*). We would like to clarify that there is **no universally optimal grouping strategy** that guarantees the best performance. The key factor determining model effectiveness is whether we can construct a sufficiently expressive **dependency injection function**, enabling accurate modeling of spatial causality. Importantly, this conclusion is tied to the specific form of dependency injection. If we move beyond simple neighbor aggregation, other grouping strategies may become equally or more effective. For instance, in our recent (unpublished) experiments, when adopting a Flamingo-style in-context aggregation mechanism—where features from the previous group are directly incorporated as context—all grouping strategies we tested achieve similarly strong performance. This suggests that **the design of the dependency injection function is more fundamental than the choice of grouping itself**.
>
> ---
>
> ### **2. How is recomputation actually hidden in practice?**
>
> #### **(a) Scheduling strategy**
>
> We describe the sampling optimization in Section 2.4 (*Autoregressive Sampling*). The recomputation overhead is **amortized within the normal sampling process**, which is why we state that the associated latency can be hidden. While there is a slight increase in total computation, the **practical latency overhead is negligible**.
>
> #### **(b) Measured overhead**
>
> We report empirical latency in Tables 1 and 2 by measuring the **average sampling time under identical hardware configurations**. Our primary focus is on **end-to-end autoregressive sampling latency**, rather than isolated metrics such as FLOPs or peak memory.
>
> ---
>
> ### **3. Can you strengthen the scaling law evidence?**
>
> Our current experiments are conducted on ImageNet, where we evaluate three model scales ranging from **141M to 1.4B parameters**, which already covers the typical range explored in prior work.
>
> However, ImageNet is a relatively simple class-conditional generation benchmark and is prone to **early saturation and overfitting**. In our experiments, models at the ~1B scale already achieve near-complete fitting, making it difficult to draw meaningful conclusions from further scaling on this dataset.
>
> To address this limitation, we have conducted additional experiments on a more challenging **text-to-image generation task**. While full quantitative results are still in progress due to computational constraints, preliminary observations show that **larger models exhibit significantly faster convergence**, as evidenced by the training loss curves (see figure below). This provides supporting evidence for the scaling behavior of our approach in more complex settings.
>
> - https://anonymous.4open.science/r/ICML26-D03D/scale-size-loss-image.png
>
> ---
>
> ### **4. How does self-token prediction perform on text (discrete data)?**
>
> At present, we have not directly applied self-token prediction to text generation. The main reason is that **standard next-token prediction already provides a strict and well-established causal structure for text**, making it highly effective.
>
> As discussed in Appendix B (*Connections and Limiting Cases*), next-token prediction can be viewed as a **special case of self-token prediction**. Therefore, our framework can be directly applied to text without modification if restricted to single-token prediction per step.
>
> Beyond natural language, we also consider more general **discrete token settings**, such as VQ-VAE image tokens. In this setting, self-token prediction with Manhattan distance grouping remains effective and even outperforms strong baselines. Specifically, using LlamaGen as a baseline and its pretrained tokenizer, we train a 1.3B model (-H) and achieve **better FID (2.13 vs. 2.34)** than the 1.4B LlamaGen-XXL on ImageNet, while requiring only **31 sampling steps compared to 256 steps**. This demonstrates that our method is **well-suited for discrete token modeling beyond continuous representations**.
>
> ---
>
> ### **5. Can you report more detailed training cost metrics?**
>
> We agree that reporting training cost is important. Below we provide additional training statistics for three model scales on the text-to-image task (note that we use a MoT architecture in this setting, which leads to slightly different memory characteristics compared to ImageNet experiments):
>
> - https://anonymous.4open.science/r/ICML26-D03D/scale-size-gpu-memory-peak.png
> - https://anonymous.4open.science/r/ICML26-D03D/scale-size-grad-norm.png
> - https://anonymous.4open.science/r/ICML26-D03D/scale-size-loss-relative-time.png
>
> We will further improve this section by including **absolute training time (e.g., GPU-hours)** and **compute estimates (FLOPs)** in the final version for a more comprehensive comparison.

---

### Official Review · Reviewer_xEp4 · 2026-03-23

**Soundness:** 3
**Presentation:** 3
**Significance:** 3
**Originality:** 3
**Overall Recommendation:** 4
**Confidence:** 4

**Summary:**

This paper proposes self-token prediction, a grouped generalization of next-token autoregression in which the model predicts an entire token group $G_t$ per causal step rather than a single token. The method is specified by three objects: an ordered partition of tokens into groups, a constructor (dependency-injection map) $h_t(G_{<t})$ that builds the causal predictive state for the current group from preceding groups, and a group-causal attention mask. Standard next-token prediction is recovered as the singleton-group special case. The paper’s central thesis is therefore not simply that one can decode several tokens at once, but that causality should be attached to a constructed groupwise predictive state, instead of being identified with one-token-at-a-time decoding.

A central technical issue is that, under grouped prediction, each group has two inconsistent roles during training. It is simultaneously a prediction target and part of the teacher-forced context for subsequent groups. The paper addresses this with a duplicated context stream vs generation stream training layout, so that all groups can be processed in one forward pass under a structured mask while preserving the intended semantics of "context” versus “generation.” At inference, it introduces cache grounding: after predicting a group from its constructed input, the KV cache is recomputed from the realized generated group rather than the surrogate constructor output, so future groups condition on realized content.

The framework is instantiated as AGARIC, a strictly spatially causal image generator. AGARIC uses Manhattan-distance grouping on a 2D visual grid, a local dependency injector over causal neighbors, group-index-step conditioning, and a flow-matching head on generation groups. The experimental headline is that AGARIC achieves competitive ImageNet generation under strict spatial causality with dramatically reduced autoregressive depth. More importantly, the paper shows that not all grouping strategies are viable. Simply shortening the autoregressive chain is insufficient, and successful grouped generation depends critically on the structure of the constructed predictive state $h_t(G_{<t})$.

The paper’s actual contribution seems strongest as a new strict grouped-causal autoregressive factorization for dense visual generation. The broader framing toward “unified multimodal autoregressive modeling” is plausible as motivation, but the paper’s evidence is primarily about causal image generation.

**Compliance With Llm Reviewing Policy:**

Affirmed.

**Final Justification:**

My final recommendation remains weak accept.

I find the paper technically interesting and potentially influential. Its main strength is the underlying idea: the work identifies the predictive-state / tokenizer construction as a central issue for grouped or continuous autoregressive generation. I also think the paper is original in the way it formulates this within a unified autoregressive framework, and the problem it addresses is clearly significant.

My main earlier concerns were about soundness and scope: whether the grouped/self-token formulation introduced an additional train–test mismatch beyond standard autoregression, and whether the broader framework/generalization claims were stronger than the paper’s current presentation and evidence justified. The rebuttal helped on both points. In particular, the clarification that dual-stream training plus cache grounding makes the grouped setting match the intended autoregressive rollout regime up to ordinary exposure bias addresses an important technical concern. Likewise, the reframing of the contribution as a more general grouped autoregressive framework, defined by token grouping plus a dependency-injection, makes the method more coherent and better motivated than the image-specific presentation alone. The mention of Flamingo-style dependency injection also makes the broader positioning more plausible.

That said, the rebuttal mainly improved the interpretation and framing of the work. I still think the strongest support is for a promising and well-motivated framework and design point. So my main reservations are only partially resolved: the paper is stronger in its clarified form, but the final version should more clearly distinguish the general abstraction from the current image-specific instantiations and should state the train–test consistency claim precisely.

Overall, the rebuttal made me somewhat more positive, but it did not materially change my overall evaluation. I remain at weak accept because I think the paper contains a real technical idea, addresses an important problem, and is likely to be useful to the community, even though some of the broader claims should be presented more carefully in the final version.

**Key Questions For Authors:**

1. What is the precise methodological distinction from recent blockwise AR+diffusion hybrids such as [2,3]?
Both already extend the autoregressive unit from tokens to blocks, use clean-past conditioning, and exploit cache-compatible causal inference. A clear response would help isolate whether the main novelty here is the self-token abstraction itself, the constructor-based predictive-state semantics, the duplicated-stream training scheme, the cache-grounding procedure, or some empirical advantage of the full recipe.
2. Do duplicated context/generation streams plus cache grounding fully resolve the train/test mismatch induced by grouped prediction, or does a gap remain relative to true self-rollout training?
This is motivated by [1] which argues that teacher-forced or partially noise-conditioned training still does not match the inference-time model distribution unless one explicitly trains on self-generated rollouts.
3. How sensitive are the conclusions to the tokenizer?
Recent works [5,6] suggest that generative performance can move dramatically even when the downstream model is fixed, and that AR and non-AR models may prefer different tokenizers or latent trajectories. Since the present paper’s strongest mechanism can also be interpreted as improving the deconstructed predictive state seen by the model, it would be useful to know how robust the conclusions are to alternative tokenizers or latent spaces.
4. Can the authors better justify the broader multimodal framing with at least one non-image experiment?
The paper currently provides strong evidence for causal image generation, but not yet for general multimodal conditional modeling. Even one modest heterogeneous conditional-generation experiment would materially strengthen the significance claim.

**Limitations:**

The paper discusses limitations, but this section should be strengthened.
In particular, the authors should state more explicitly that:
- the empirical evidence is primarily about class-conditional image generation, not general multimodal conditional modeling;
- the most important design choices, i.e. the grouping and the constructor $h_t(\cdot)$, are currently hand-designed and modality-specific, and principled design rules for new modalities are still lacking.
- closely related prior work already exists on blockwise AR plus diffusion hybrids, causal objectives beyond literal immediate NTP, collective structured prediction, so the novelty should be stated more narrowly and precisely;
- the broader hypothesis that strict spatial causality may be important for future multimodal capability remains speculative.

**Strengths And Weaknesses:**

Overall, this is a technically interesting paper. Its strongest contribution is narrower than some of the framing suggests, but at the right level, it is substantial: the paper proposes a strict grouped-causal autoregressive factorization in which grouping, constructor-defined predictive states, duplicated-stream training, and inference-time cache grounding jointly define the semantics of generation.

Strengths:
1. The paper identifies that grouped causal generation is fundamentally a predictive-state construction problem. The technical strength is that the paper does not treat grouped generation as a superficial decoding-speed trick. Once one departs from shift-aligned next-token supervision, the key problem becomes how to construct a causal state that is both computable from the past and semantically aligned with future prediction. The paper gives a coherent answer to this.

2. The duplicated context/generation streams and cache grounding are technically meaningful.
A major virtue is that the paper explicitly recognizes the semantic inconsistency induced by grouped prediction. The current group is predicted from a constructed surrogate state, but future groups should condition on the realized generated group. The duplicated-stream construction and inference-time cache grounding are principled responses to that mismatch. A useful nearby reference point is [1], which emphasizes that autoregressive diffusion models suffer from a train/test distribution gap because they are trained on ground-truth or partially noised context but rolled out on self-generated context. Relative to that literature, a genuine strength here is that the paper explicitly formulates grouped generation as a causal-state consistency problem.

3. The experimental section is strong for the grouping/dependency ablation that directly tests the central mechanism. It asks whether gains come from simply shortening the autoregressive chain, or from constructing a better causal predictive state. The results strongly support the latter. Row-wise and next-layer grouping sharply reduce depth and latency but degrade sample quality, while Manhattan grouping with a meaningful local constructor preserves strong quality. Moreover, replacing a non-informative constructor with neighborhood aggregation produces a large FID improvement while holding the grouped setup fixed. This provides strong support for the paper’s central claim, because it shows that the decisive object is not “multi-token prediction” but the structured causal state used to predict the current group. The optimization observations reinforce the same point: with poor grouping or constructor choices, convergence can remain catastrophically bad for a long time.

4. The guidance analysis is mechanism-matched rather than generic tuning. The paper argues that under strict spatial causality, the unconditional branch becomes increasingly informative as generated context accumulates, weakening standard CFG. The cosine-similarity analysis and guidance ablation support that story.

5. The paper contributes something original, but only at the right level of abstraction. There is now a substantial nearby literature challenging the naive one-token bottleneck, but mostly in adjacent regimes [1-4].  The submission contributes a distinct strict grouped-causal autoregressive abstraction viewpoint in which the grouping and constructor define the predictive state itself, together with explicit train/inference semantics for that state.

Weaknesses

1. The broadest framing outruns the evidence.
The paper repeatedly motivates itself through “unified and efficient multimodal autoregressive modeling,” but nearly all substantive evidence is on class-conditional ImageNet image generation.

2. The related-work positioning should be sharper and more technically precise.
The paper is strongest when it claims novelty at the level of the self-token factorization and its semantics. Recent blockwise AR+diffusion hybrids such as [2] and [3] already occupy a nearby design region, while [1] and [4] already show that causal generation need not equal literal immediate next-token supervision. This does not eliminate the paper’s novelty, but it narrows it considerably. The paper would be stronger if it stated this more clearly.

3. The experiments do not fully isolate the contribution of the self-token primitive from the rest of the AGARIC stack.
The best system combines grouped prediction, Manhattan grouping, a hand-designed local constructor, duplicated-stream training, cache grounding, group-index conditioning, local noise injection, a flow-matching head, and modified CFG. The paper usefully ablates several of these components, but it still does not provide a single clean decomposition from a strong next-token causal baseline to the full system.

4. It remains unclear whether duplicated-stream training plus cache grounding fully closes the grouped train/test gap.
[1] makes a stronger point that unless training explicitly optimizes on self-generated rollouts drawn from the inference-time model distribution, there can remain a gap between the causal states seen in training and those encountered at rollout. The present paper partially addresses this through duplicated semantics and cache grounding, but it is not fully clear that this closes the gap in the stronger Self-Forcing sense. Given how central the grouped train/test mismatch is to the method, the experimental support on this point feels thinner than the conceptual importance of the claim.

5. The efficiency story is plausible but not fully standardized.
The paper clearly reduces autoregressive depth and likely reduces latency relative to tokenwise AR, and the latency table is useful. However, the practical case would be much stronger with a more explicit accounting of the cost of cache grounding, since grounding is central to the method’s semantics. I would have liked a direct latency/memory ablation with and without grounding, or at least a clearer forward-pass-equivalent accounting. As written, the efficiency story is persuasive but not maximally rigorous.

6. The representation-universality claim is somewhat overstated.
The representation study is useful, but it does not yet justify any strong claim that tokenizers no longer matter. The raw-pixel result remains materially worse than the best latent-space result, and the raw setting also changes the prediction parameterization. This matters especially because nearby work shows that tokenizer/interface design can itself dramatically move the generative frontier. A fair interpretation is therefore that some of the gains may come from a better predictive or reconstruction interface, not only from grouped causality alone. This reading is reinforced by [5-6], all of which emphasize that generation quality depends heavily on the order and structure in which information is presented to the model.

7. Guidance is doing a large amount of work.
The paper is very transparent about this, and this is more a framing point. No-guidance performance is very poor; standard CFG already provides the dominant jump; x-CFG/z-CFG then refine further. So one should be careful not to interpret that the bare self-token factorization itself is already near diffusion-quality generation, but requires non-trivial guidance engineering.

Overall:

Soundness: good overall. The grouped-causal semantics are coherent, and the mechanism-level experiments are well chosen. The main remaining soundness issue is whether duplicated-stream training plus cache grounding fully resolves the grouped train/test mismatch, and whether the final gains can be cleanly attributed to the self-token primitive itself.

Presentation: generally strong, but the claims should be scoped more carefully and the related-work positioning should be sharpened. In particular, the paper should distinguish more explicitly between what it establishes for causal image generation and what remains speculative for multimodal modeling.

Significance: high for causal visual generation. The broader multimodal significance is currently more aspirational than experimentally demonstrated.

Originality: strong at the level of strict grouped-causal autoregressive factorization with explicit predictive-state construction and train/inference semantics. The originality claim becomes much weaker if phrased more broadly as “multi-token causal generation”.

References:\
[1] Huang, Xun, et al. "Self forcing: Bridging the train-test gap in autoregressive video diffusion." arXiv preprint arXiv:2506.08009 (2025). \
[2] Hu, Jinyi, et al. "Acdit: Interpolating autoregressive conditional modeling and diffusion transformer." arXiv preprint arXiv:2412.07720 (2024).\
[3] Chen, Junhao, Yulia Tsvetkov, and Xiaochuang Han. "MADFormer: Mixed Autoregressive and Diffusion Transformers for Continuous Image Generation." arXiv preprint arXiv:2506.07999 (2025). \
[4] Yang, Shu-wen, et al. "Generative audio language modeling with continuous-valued tokens and masked next-token prediction." arXiv preprint arXiv:2507.09834 (2025).\
[5] Yang, Jiawei, et al. "Latent denoising makes good visual tokenizers." arXiv preprint arXiv:2507.15856 (2025).\
[6] Yu, Lijun, et al. "Language Model Beats Diffusion--Tokenizer is Key to Visual Generation." arXiv preprint arXiv:2310.05737 (2023).

---

> ### Author Rebuttal · Authors · 2026-03-30
>
> ## 1. What is the *true core novelty* of the method?
>
> The core contribution of **self-token prediction** is to challenge the prevailing assumption induced by next-token prediction (NTP)—namely, that autoregressive models must generate **one token at a time**. This assumption has shaped many existing autoregressive approaches for image and video generation, leading to prohibitively high latency when scaling to high-resolution settings.
>
> Self-token prediction removes this constraint by allowing **arbitrary grouping of tokens**, enabling the model to predict multiple tokens per step. More importantly, our work shows that the key to effective autoregressive modeling is **not the granularity of prediction**, but rather **how the predictive state is constructed via dependency injection**. In particular, designing an effective dependency injection function that captures **spatial causality** is the primary determinant of generation quality.
>
> The dual-stream design is introduced to address a practical issue: in the most general grouped setting—where token groups and dependency injection are unconstrained—it is non-trivial to train all groups efficiently in parallel. The dual-stream formulation provides a principled way to enable such parallel training. However, we emphasize that **it is not inherently required by self-token prediction** (see Appendix D: *More Empirical Studies*).
>
> ---
>
> ## 2. Does the method *fully resolve the train–test mismatch*?
>
> With the introduction of **dual-stream training** and **cache grounding during inference**, our framework ensures **theoretical consistency between training and inference**, as illustrated in Figures 2 and 3.
>
> However, we would like to clarify an important point:
> **self-token prediction itself is inherently train–test consistent**. The mismatch discussed in Section 2 arises only when we aim to support **fully parallel training over all token groups** in the most general formulation. In this case, dual-stream training is required to maintain correct semantics during parallelization.
>
> ---
>
> ## 3. Where do the *performance gains actually come from*?
>
> As shown in Table 1, the primary factor behind the success of self-token prediction is the ability to construct an **effective spatial causal structure**. There is no universally optimal grouping strategy; instead, performance depends on whether a given grouping allows for a **meaningful and expressive dependency structure** that supports learning.
>
> Beyond quality, an important advantage of self-token prediction lies in **latency reduction**. By predicting multiple tokens per step, it significantly shortens the autoregressive chain. In contrast, next-token prediction suffers from inherently long generation times for high-resolution images and videos. Self-token prediction addresses this limitation at the modeling level, rather than through engineering optimizations.
>
> ---
>
> ## 4. What is the *real efficiency benefit*?
>
> Cache grounding introduces additional computation, as the KV cache needs to be recomputed based on realized tokens. In the worst case, this can approximately double the compute relative to naive next-token prediction.
>
> However, two important clarifications are needed:
>
> * This overhead arises **only when dual-stream training is used**. If the dependency injection function is designed differently (e.g., using an in-context formulation similar to Flamingo), both dual-stream training and cache grounding can be avoided.
> * In practice, we observe that the cost of cache recomputation can be **effectively overlapped with the sampling of the next token group**, and does not require additional forward passes. As shown in Figure 3, this results in **minimal impact on overall latency**.
>
> Overall, the efficiency gains of self-token prediction come primarily from **reducing the number of autoregressive steps via grouped prediction**, rather than reducing total FLOPs.
>
> ---
>
> ## 5. How *general and robust* is the approach?
>
> We provide evidence of generality in Figures 10 and 16, where self-token prediction is applied to **video generation and understanding tasks**. In addition, we explore alternative dependency injection designs, such as a Flamingo-style in-context mechanism for text-to-image generation:
>
> * https://anonymous.4open.science/r/ICML26-D03D/avg-vs-flamingo-loss-curve.png
>
> Replacing local aggregation with this more general mechanism leads to **faster convergence and reduced reliance on manual design**, suggesting that the framework is flexible and extensible.
>
> In the main paper, we focus on local aggregation to better highlight the importance of **causality in autoregressive modeling**. However, self-token prediction is **not restricted to local aggregation**, nor to continuous latent spaces. We have also conducted experiments with **discrete image tokenization**, which similarly demonstrate the effectiveness of the approach (see response to Reviewer Hkkz, Q4).

---

> > ### Author Rebuttal · Reviewer_xEp4 · 2026-04-03
> >
> > Thank you for the rebuttal. It helps clarify that the paper’s core novelty is showing that grouped autoregression succeeds or fails based on the quality of the constructed predictive state. This usefully sharpens the originality claim.
> >
> > However, my main concerns are only partially resolved.
> >
> > First, on train–test mismatch, the rebuttal clarifies semantic consistency of the grouped formulation and why dual-stream training/cache grounding are introduced. But my original concern was stronger: whether the distribution of grouped causal states seen during training is sufficiently aligned with the one induced by actual inference-time rollout. I understand the claim of formal consistency of the factorization, but I am still not fully convinced this resolves the stronger rollout-distribution issue.
> >
> > Second, on generality, the extra Flamingo-style dependency-injection result is promising and partially addresses my concern that the method might be too tied to hand-designed local image aggregation. Still, I think the paper’s broader multimodal/general-framework framing remains only partially supported by the evidence currently in the main paper.
> >
> > Follow-up questions
> > 1. Your rebuttal says dual-stream training with cache grounding ensure “theoretical consistency.” Is this a claim that the implemented computation matches the intended grouped factorization $p_\theta(G_t \mid h_t(G_{<t}))$, or a stronger claim that the training objective optimizes under a state visitation distribution close to the one induced by inference-time rollout? My concern is mainly about the latter, since semantic correctness alone does not rule out grouped exposure bias.
> >
> > 2. Do you view the Flamingo-style result as evidence that the main contribution is best understood as a general predictive-state construction framework, with local aggregation only one image-specific instantiation? If so, I think making that explicit in the paper would significantly improve the presentation and claim calibration.

---

> > > ### Author Response · Authors · 2026-04-06
> > >
> > > ### **Response to Train–Test Mismatch Concern**
> > >
> > > We thank the reviewer for highlighting the important distinction between *factorization correctness* and *rollout distribution alignment*. We would like to clarify that our claim of “theoretical consistency” refers to a stronger notion than merely matching the computation graph.
> > >
> > > Specifically, in self-token prediction with dual-stream training and cache grounding, **the distribution of context tokens observed by the model at each step (t) is identical between training and inference**. At any step, the model conditions on the realized tokens from all preceding groups, and these tokens enter the KV cache in exactly the same way during both training and inference.
> > >
> > > The only remaining difference is the standard one inherent to all autoregressive models:
> > >
> > > * during training, preceding tokens come from the data distribution;
> > > * during inference, they are generated by the model itself.
> > >
> > > Importantly, this distinction does **not introduce an additional mismatch specific to grouped prediction**, but is instead the same exposure bias present in standard next-token autoregression. In other words, self-token prediction with cache grounding reduces the grouped setting to the *same train–test regime as conventional autoregressive models*, rather than introducing a new source of mismatch.
> > >
> > > Therefore, our method ensures not only semantic correctness of the factorization, but also that **the state distribution seen during training matches the intended autoregressive rollout distribution up to the standard autoregressive assumption**. We will clarify this distinction explicitly in the revision.
> > >
> > > ---
> > >
> > > ### **Response to Generality and Framework Positioning**
> > >
> > > We agree with the reviewer that the current presentation may over-emphasize the image-specific instantiation.
> > >
> > > Our primary contribution is best understood as a **general framework for grouped autoregressive modeling**, where generation is defined by two key components:
> > > (1) a grouping of tokens, and
> > > (2) a constructor (dependency injection function) that defines the predictive state.
> > >
> > > Under this view, the Manhattan grouping and local aggregation used in ImageNet experiments are **one particular instantiation**, designed to validate that effective grouped autoregression is achievable in a strict causal setting. We agree that these choices are largely modality-specific.
> > >
> > > Importantly, our additional experiments with **Flamingo-style dependency injection** suggest that the framework is not tied to hand-designed local aggregation. This alternative design can be applied with minimal modification across modalities and shows improved optimization behavior, supporting the interpretation of self-token prediction as a **general predictive-state construction paradigm** rather than an image-specific technique.
> > >
> > > In the final version, we will revise the paper to:
> > >
> > > * make this abstraction explicit,
> > > * better separate the general framework from specific design choices, and
> > > * incorporate the Flamingo-style results into the main discussion to more directly support the multimodal claim.

---

### Decision · Program_Chairs · 2026-04-30

**Decision:**

Accept (regular)

**Comment:**

The propose self-token prediction as a grouped autoregressive framework. The reviewers find the idea interesting and supported by solid experiments, they raise major concerns about claim of the paper being to broad. Therefore we recommand acceptance but invite authors to incorporate reviewers feedback in the final version.